# Fréchet Wavelet Distance: A Domain-Agnostic Metric for Image Generation

**Lokesh Veeramacheneni**
University of Bonn
*lveerama@uni-bonn.de*

**Moritz Wolter**
University of Bonn
*moritz.wolter@uni-bonn.de*

**Hildegard Kuehne**
University of Tuebingen,
MIT-IBM Watson AI Lab
*h.kuehne@uni-tuebingen.de*

**Juergen Gall**
University of Bonn,
Lamarr Institute for Machine Learning and Artificial Intelligence
*gall@iai.uni-bonn.de*

## Abstract

Modern metrics for generative learning like Fréchet Inception Distance (FID) and DINOv2-Fréchet Distance (FD-DINOv2) demonstrate impressive performance. However, they suffer from various shortcomings, like a bias towards specific generators and datasets. To address this problem, we propose the Fréchet Wavelet Distance (FWD) as a domain-agnostic metric based on the Wavelet Packet Transform ($\mathcal{W}_p$). FWD provides a sight across a broad spectrum of frequencies in images with a high resolution, preserving both spatial and textural aspects. Specifically, we use $\mathcal{W}_p$ to project generated and real images to the packet coefficient space. We then compute the Fréchet distance with the resultant coefficients to evaluate the quality of a generator. This metric is general-purpose and dataset-domain agnostic, as it does not rely on any pre-trained network, while being more interpretable due to its ability to compute Fréchet distance per packet, enhancing transparency. We conclude with an extensive evaluation of a wide variety of generators across various datasets that the proposed FWD can generalize and improve robustness to domain shifts and various corruptions compared to other metrics.

## 1 Introduction

With the surge of generative neural networks, especially in the image domain, it becomes important to assess their performance in a robust and reliable way (Heusel et al., 2017a; Binkowski et al., 2018; Salimans et al., 2016; Kynkäänniemi et al., 2019; Stein et al., 2023). FID (Heusel et al., 2017a) has emerged as the de facto standard for comparing generative image synthesis approaches. However, it also shows various shortcomings, such as its reliance on a pre-trained classification backbone, i.e., InceptionV3 trained on ImageNet. This, by design, introduces a class dependency into FID leading to accidental distortions (Sauer et al., 2021). The FID scores improve if the evaluation set resembles ImageNet or if the use of an ImageNet pre-trained discriminator pushes the output distribution towards ImageNet, although the image quality remains the same in these cases (Kynkäänniemi et al., 2023).

To address the domain bias problem caused by the use of a pre-trained network, we propose an alternative metric based on the Wavelet Packet Transform ($\mathcal{W}_p$). In contrast to other pure frequency (Narwaria et al., 2012) or spatial (Wang et al., 2004; Horé & Ziou, 2010) metrics, wavelets have the advantage that they combine both frequency and spatial aspects in one metric. While frequency information is important (Durall et al., 2020; Dzanic et al., 2020; Rahaman et al., 2019; Schwarz et al., 2021; Wolter et al., 2022), it alone is insufficient to assess the quality of synthesized images without considering additional spatial information. Wavelets are thus an ideal representation for a metric comparing generative approaches for image synthesis. As FID, FWD utilizes the Fréchet distance of the real and generated set of images as a distance measure, but it is not computed based on InceptionV3 activation maps. Instead, it utilizes the wavelet-packet frequency band representations of $\mathcal{W}_p$ as illustrated in Figures 1 and 3. To this end, we first use $\mathcal{W}_p$ to transform every image, where

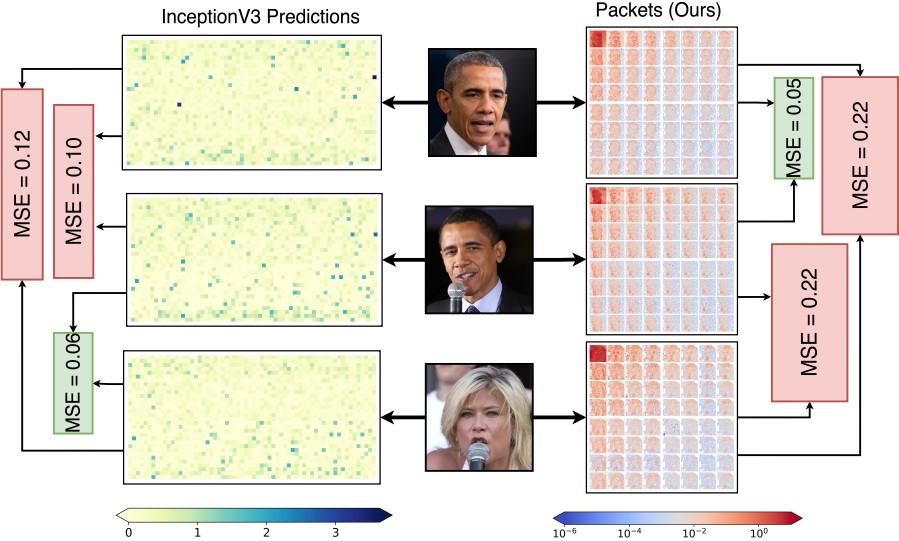

Figure 1: The first two images depict the same person, while the last image depicts a different person. Intuitively, the first two images are more similar than the other pairs of images. When computing the mean squared error between the images using the penultimate InceptionV3 activations or wavelet packets, we observe that the wavelet packets produce a low distance for the first two images, as expected. Surprisingly, according to InceptionV3, the last two images are similar since both images are classified as 'microphone' whereas the first image as 'groom'. Images from Flickr.

we use the Haar wavelet transform at a fixed level. We then compute the Fréchet distance for each packet of the transform and average them over all packets. The proposed Fréchet Wavelet Distance (FWD) thus considers spatial information as well as all frequency bands.

To quantitatively assess those characteristics, we evaluate the proposed metric in terms of its domain bias and robustness. We further compare the proposed FWD to existing state-of-the-art metrics like FID, Kernel Inception Distance (KID), and DINOv2-Fréchet Distance (FD-DINOv2) on standard datasets. We show that FWD is a more robust metric that does not suffer from the domain bias and can thus be applied to any dataset. Kynkäänniemi et al. (2023) experimented with optimizing FID by selecting a subset of images from 250k generated images, where the subset's InceptionV3 activations are related to ImageNet classes. Building on this work, we observed a significant improvement in FID by $\approx 50\%$, when evaluated on this subset. FD-DINOv2 responded to ImageNet-feature optimization with an improvement of $\approx 2\%$ as well. This undesired improvement can likely be explained by the overlap between the ImageNet and the DINOv2 training set. In contrast, FWD remains the same despite the manipulation. We also show that some unexpected FID results can be attributed to the dataset bias. Furthermore, FWD is significantly faster to compute. The source code for computing FWD is available at: https://github.com/BonnBytes/PyTorch-FWD.

In summary, this paper makes the following contributions:

1. We propose the Fréchet Wavelet Distance (FWD) as a dataset- and domain-agnostic metric for evaluation of generative approaches for image synthesis.

2. FWD is an interpretable metric, as the Wavelet Packet Transform ($\mathcal{W}_p$) splits the frequency space into hierarchically organized, discrete subbands.

3. We show that the proposed method is computationally inexpensive and robust to corruption, perturbation, and distractors.

4. We show that FD-DINOv2 addresses the domain bias issue to an extent but at a very high computational cost. Furthermore, we provide evidence that it is still limited to its training data domain.

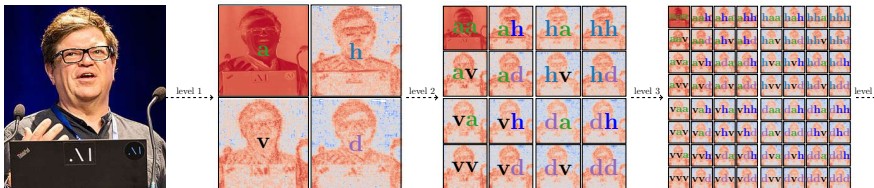

Figure 2: Illustration of the Wavelet Packet Transform ($\mathcal{W}_p$). For visualization purposes, we depict a level-3 transform. All later experiments use a level-4 transform. Image from Jérémy Barande (2024).

## 2    RELATED WORK

### 2.1    METRICS FOR GENERATIVE LEARNING

A generative model should generate novel image samples that mirror the training set sample distribution, including data diversity. In a vision context, Salimans et al. (2016) proposed the Inception Score (IS) as a measure of image quality, independent of the target dataset statistics. The IS is computed by measuring the entropy of the class probabilities of an InceptionV3. The score builds upon the assumption that a generative network that has converged to a meaningful solution will produce images that will allow InceptionV3 to make predictions with certainty. In other words, a certain InceptionV3 has a low prediction entropy. IS has been found to be sensitive to different ImageNet training runs (Barratt & Sharma, 2018). Furthermore, it does not use the statistics of the real data distribution a Generative Adversarial neural Network (GAN) is trained to model (Heusel et al., 2017a). Heusel et al. (2017a) proposed FID in response. Instead of measuring the entropy at the final layer, FID is computed by evaluating the Fréchet distance (Dowson & Landau, 1982) between the penultimate network activations computed on both the true and synthetic images. Today, comparing high-level InceptionV3 features using an FID-score (Heusel et al., 2017a) enjoys widespread adoption and several variants exist. Kernel Inception Distance (KID) (Binkowski et al., 2018), for example, relaxes the multivariate Gaussian assumption of FID and measures the polynomial kernel distance between Inception features of the generated and the training dataset. Binkowski et al. (2018) kept the InceptionV3 backbone and replaced the Fréchet distance with a kernel distance. While FID captures general trends well, the literature also discusses its drawbacks. Kynkäänniemi et al. (2023) empirically studied the effect of ImageNet classes on FID for non-ImageNet datasets by using GradCAM. Furthermore, Kynkäänniemi et al. (2023) examined ImageNet bias using Projected Fast GAN (Proj. FastGAN) and StyleGAN2. Compared to StyleGAN2, Proj. FastGAN produces more accidental distortions like floating heads and artifacts (Sauer et al., 2021). Surprisingly, Proj. FastGAN's FID is comparable to StyleGAN2's in their experiment. Chong & Forsyth (2020) found a generator-dependent architecture bias, which limits the ability to compare samples for smaller datasets with 50K or fewer images. Additionally, Parmar et al. (2022) found that both FID and KID are highly sensitive to resizing and compression. Barratt & Sharma (2018) reported FID sensitivity with respect to different InceptionV3 weights. While comparing Tensorflow and PyTorch implementations, Parmar et al. (2022) measured inconsistent scores due to differing resizing implementations. Finally, FID scores are hard to reproduce unless all details regarding its computation are carefully disclosed (Hug, 2024). Stein et al. (2023) proposed an alternative to over-reliance on InceptionV3, by replacing it with the DINOv2-ViT-L/14 model (Oquab et al., 2024). This replacement partially addresses the domain bias problem but significantly increases computational cost. Unfortunately, DINOv2's training dataset is not publicly available. Furthermore, existing frequency-based metrics such as Sliced Wasserstein Distance (SWD) proposed in Karras et al. (2018) involves multiple projections on a random basis. In spite of its ability to detect domain bias, it suffers from reproducibility issues (Nguyen et al., 2023) due to random projections. Consequently, gaps in the dataset remain hidden. This situation motivates the search for additional quality metrics. A detailed discussion of spectral methods and generative architectures is presented in supplementary Section B.

## 3    FRÉCHET WAVELET DISTANCE (FWD)

We want to tackle the problem of dataset-domain bias. To this end, we propose FWD, which in turn leverages the Wavelet Packet Transform ($\mathcal{W}_p$). We require two-dimensional filters for image

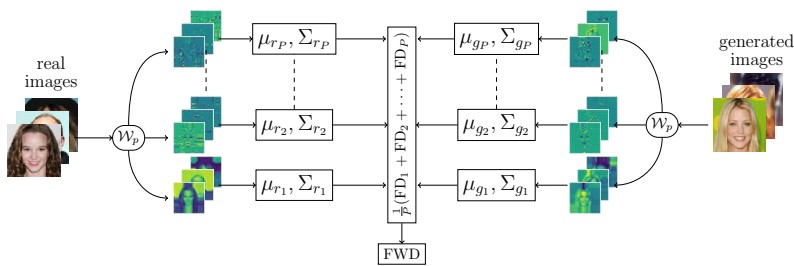

Figure 3: Fréchet Wavelet Distance (FWD) computation flow-chart. $\mathcal{W}_p$ denotes the wavelet-packet transform. Not all packet coefficients are shown, dashed lines indicate omissions. We compute individual Fréchet Distances for each packet coefficient and finally average across all the coefficients.

processing. We start with single-dimensional Haar wavelets. Next, we construct filter quadruples from the original single-dimensional filter pairs. The process uses outer products (Vyas et al., 2018):

$$\mathbf{h}_a = \mathbf{h}_{\mathcal{L}}\mathbf{h}_{\mathcal{L}}^T, \mathbf{h}_h = \mathbf{h}_{\mathcal{L}}\mathbf{h}_{\mathcal{H}}^T, \mathbf{h}_v = \mathbf{h}_{\mathcal{H}}\mathbf{h}_{\mathcal{L}}^T, \mathbf{h}_d = \mathbf{h}_{\mathcal{H}}\mathbf{h}_{\mathcal{H}}^T, \tag{1}$$

with $a$ for the approximation filter, $h$ for the horizontal filter, $v$ for the vertical filter, and $d$ for the diagonal filter (Lee et al., 2019). We construct a $\mathcal{W}_p$-tree for images with these two-dimensional filters, as illustrated in Fig. 2, using recursive convolution operations with the filter quadruples, i.e.,

$$\mathbf{C}_{\mathcal{F}_l} * \mathbf{h}_j = \mathbf{C}_{\mathcal{F}_{l+1}}, \tag{2}$$

at every recursion step where $*$ denotes a two-dimensional convolution with a stride of two. The filter codes $\mathcal{F}_{l+1}$ are constructed by applying all $j \in [a, h, v, d]$ filters to the previous filter codes $\mathcal{F}_l$. Initially, the set of inputs $F_l$ will only contain the original image $\mathbf{C}_{F_0} = \{X\}$ as shown in Fig. 2. At level one, we obtain the result of all four convolutions with the input image and have $F_1 = [a, h, v, d]$. At level two, we repeat the process for all elements in $F_1$. $F_2$ now contains two-character keys $[aa, ah, av, ad, \ldots, dv, dd]$ as illustrated in Fig. 2. We typically continue this process until level 4 in this paper. We arrange the coefficients in $\mathbf{C}_{\mathcal{F}_l}$ as tensors $\mathbf{C}_l \in \mathbb{R}^{P, H_p, W_p}$ for the final layer. The total number of packages at every level is given by $P = 4^l$, $H_p = \frac{H}{4^l}$ and $W_p = \frac{W}{4^l}$, where we denote the image height and width as $H$ and $W$. We provide more details on $\mathcal{W}_p$ in the Supplementary.

Figure 3 illustrates how we compute the FWD. The process relies on the wavelet packet transform, as previously discussed. We process $N$ images with $C$ channels in parallel $\mathcal{W}_p : I_s \in \mathbb{R}^{N \times H \times W \times C} \rightarrow \mathbf{C} \in \mathbb{R}^{N \times P \times H_p \cdot W_p \cdot C}$, where $H$ and $W$ denote image height and width as before. To facilitate the ensuing metric evaluation, we flatten the last axes into $(H_p \cdot W_p \cdot C)$. Before computing the packets, all pixels are divided by 255 to re-scale all values to [0,1]. The metric is computed in three steps. First, we compute the individual packet mean via

$$\mu_p(I_N) = \frac{1}{N} \sum_{n=1}^N \mathcal{W}(I_n)_p, \tag{3}$$

where $I_n$ is the $n^{th}$ image in the dataset and $p$ represent the corresponding packet from $P$ packets. Then we compute the covariance matrix as

$$\Sigma_p(I_N) = \frac{1}{N-1} \sum_{n=1}^N (\mathcal{W}(I_n)_p - \mu_p(I_N))(\mathcal{W}(I_N)_p - \mu_p(I_N))^T. \tag{4}$$

Here, $\mu \in \mathbb{R}^{P \times C \cdot H_p \cdot W_p}$ denotes the mean across the number of images, and $\Sigma \in \mathbb{R}^{P \times C \cdot H_p \cdot W_p \times C \cdot H_p \cdot W_p}$ represents the covariance among all the coefficients. Now we are ready to compute the distances given the packet mean and covariance values,

$$\text{FD}_p(r, g) = d(\mathcal{N}(\mu_{r_p}, \Sigma_{r_p}), \mathcal{N}(\mu_{g_p}, \Sigma_{g_p}))^2 = ||\mu_{r_p} - \mu_{g_p}||_2^2 + \text{Tr}[\Sigma_{r_p} + \Sigma_{g_p} - 2\sqrt{\Sigma_{r_p}\Sigma_{g_p}}], \tag{5}$$

with $r$ and $g$ denoting the real and generated images and Tr denoting the trace operation. Utilising the above computed per-packet statistics for both real $(\mu_r, \Sigma_g)$ and generated samples $(\mu_r, \Sigma_g)$, we measure the mean of Fréchet Distance (Equation 5) across all packets

$$\text{FWD} = \frac{1}{P} \sum_{p=1}^P d(\mathcal{N}(\mu_{r_p}, \Sigma_{r_p}), \mathcal{N}(\mu_{g_p}, \Sigma_{g_p}))^2. \tag{6}$$

Table 1: Comparison of FID, FD-DINOv2 and FWD to depict domain bias. FID prefers Proj. FastGAN over Denoising Diffusion GAN (DDGAN) across all the datasets. Whereas FWD prefers Denoising Diffusion GAN (DDGAN). We find that FD-DINOv2 agrees with FWD across all datasets except Deep Nutrient Deficiency Dikopshof Dataset (DNDD-Dataset). This might be because agriculture data is not part of DINOv2's training set.

| Dataset | Generator | FID↓ | FD-DINOv2↓ | FWD↓ (ours) |
|---|---|---|---|---|
| CelebA-HQ | Proj. FastGAN | **6.358** | 685.889 | 1.388 |
| | DDGAN | 7.641 | **199.761** | **0.408** |
| FFHQ | Proj. FastGAN | **4.106** | 593.124 | 0.651 |
| | StyleGAN2 | 4.282 | **420.273** | **0.312** |
| DNDD-Dataset | Proj. FastGAN | **4.675** | **171.625** | 1.442 |
| | DDGAN | 26.233 | 232.884 | **1.357** |
| Sentinel | Proj. FastGAN | **8.96** | 424.898 | 0.755 |
| | DDGAN | 23.615 | **404.700** | **0.115** |

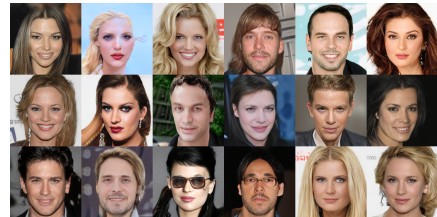

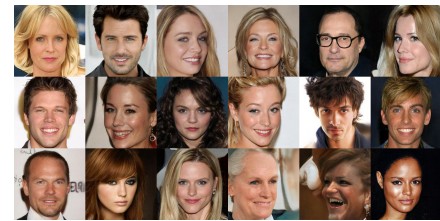

(a) Proj. FastGAN on CelebA-HQ (FID: 6.358, FWD: 1.388)

(b) DDGAN on CelebA-HQ (FID: 7.641, FWD: 0.408)

Figure 4: Samples from (a) Proj. FastGAN and (b) DDGAN on the Large-scale Celeb Faces Attributes High Quality (CelebA-HQ) dataset. The FID prefers Proj. FastGAN irrespective of visual artefacts and floating heads, whereas our metric (FWD) ranks DDGAN higher than Proj. FastGAN.

By averaging the distances of all frequency bands, the FWD captures frequency information across the spectrum.

## 4 EXPERIMENTS

Our first series of experiments demonstrates the effect of domain bias on learned metrics, demonstrating the resilience of FWD to such bias. All experiments were implemented using the same code base. **Implementation:** We use PyTorch (Paszke et al., 2017) for neural network training and evaluation and compute FID using (Seitzer, 2020) as recommended by Heusel et al. (2017b). We work with the wavelet filter coefficients provided by PyWavelets (Lee et al., 2019). We chose the PyTorch-Wavelet-Toolbox (Wolter et al., 2024) software package for GPU support. FD-DINOv2 and KID are computed using the codebases from Stein et al. (2023) and Binkowski et al. (2018), respectively.

### 4.1 EFFECT OF DOMAIN BIAS

Kynkäänniemi et al. (2023) observed that metrics based on ImageNet-trained network features emphasize ImageNet-related information. This behaviour is desired when we evaluate generators on ImageNet or similar datasets. When working with other datasets, this behaviour is misleading. **Datasets:** As datasets, we use Large-scale Celeb Faces Attributes High Quality (CelebA-HQ) (Karras et al., 2018), Flickr Faces High Quality (FFHQ), DNDD-Dataset (Yi et al., 2023), an agricultural dataset, and Sentinel (Schmitt et al., 2019), a remote sensing dataset. These datasets contain images that are very different from those in ImageNet. More information about the DNDD-Dataset and the Sentinel dataset can be found in the supplementary material. **Generators:** We study dataset domain bias effects using the Denoising Diffusion GAN (DDGAN),

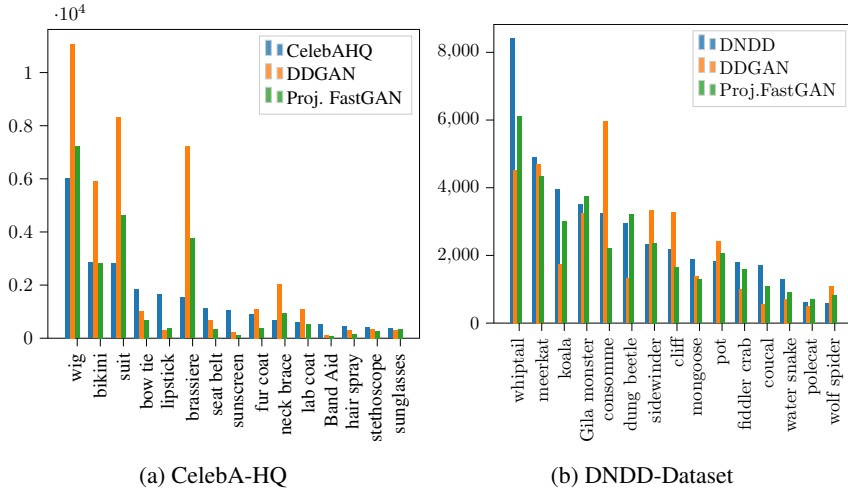

(a) CelebA-HQ                    (b) DNDD-Dataset

Figure 5: Distribution of ImageNet Top-1 classes, predicted by InceptionV3 for real images and images generated by DDGAN and Proj. FastGAN. (a) depicts the distribution for the CelebA-HQ dataset and (b) shows the distribution for DNDD-Dataset. Although irrelevant for visual quality, the class distribution of Proj. FastGAN aligns more closely with the real distribution than DDGAN for both the datasets, contributing to lower FID for Proj. FastGAN.

Proj. FastGAN and StyleGAN2 networks. Proj. FastGAN is particularly interesting since its discriminator relies on ImageNet weights to improve training convergence (Sauer et al., 2021). Prior work found this architecture to improve FID on image datasets far from ImageNet, without substantially improving image quality (Kynkäänniemi et al., 2023).

**Hyperparameters:** To examine the effect of dataset bias, we require generators, which are tuned to produce output that resembles our datasets' distribution. Specifically, we trained Proj. FastGAN for 100 epochs on both the CelebA-HQ dataset and DNDD-Dataset, respectively, using a learning rate of 1e-4 and batch size of 64 with 8 A100 GPUs. For the Sentinel dataset, we trained Proj. FastGAN for 150 epochs, using the same hardware and hyperparameters. For FFHQ, pre-trained weights are available, as well as pre-trained weights for DDGAN on CelebA-HQ from Xiao et al. (2022). On DNDD-Dataset, we trained DDGAN for 150 epochs with a learning rate of 1e-4 and batch size of 8 on the same hardware. We also trained DDGAN on the Sentinel dataset for 250 epochs, using a learning rate of 1e-4 and batch size of 4 on 4 A100 GPUs. For StyleGAN2, we use the pretrained weights with the code from Karras et al. (2020).

**Results:** Table 1 presents the FID, KID, FD-DINOv2 and FWD values across all datasets for the afore-mentioned generators. Across all datasets, FID prefers images generated by Proj. FastGAN. When we compare images generated by Proj. FastGAN and DDGAN for CelebA-HQ, which are shown in Figures 4a and 4b, we observe that more deformations are visible in the images of Proj. FastGAN compared to DDGAN images. DDGAN, in other words, produces more high-quality images. Supplementary Figures 16 and 17 illustrate this observation further. Consequently, it is surprising to see FID preferring Proj. FastGAN, as we would expect DDGAN to come out on top. Following Kynkäänniemi et al. (2023), we compare the InceptionV3 output label distribution of the original-CelebA-HQ images to their synthetic counterparts from DDGAN and Proj. FastGAN in Figure 5a. We observe that InveptionV3 produces a label distribution for Proj. FastGAN, which resembles the distribution from InveptionV3 for the original CelebA-HQ images. The label distribution for images from DDGAN differs significantly. This discrepancy, also reported by Kynkäänniemi et al. (2023), explains why FID produces a misleading verdict. FWD, in contrast, prefers DDGAN, as we would expect.

The same pattern repeats in the results for our FFHQ-experiments. Generally, we see FID preferring Proj. FastGAN images, while FWD puts StyleGAN2 on top. Our observations confirm the experiment in Kynkäänniemi et al. (2023). In a next step, we study the effect of a larger network backbone for the neural Fréchet distance computations. Stein et al. (2023) proposed to replace InveptionV3

Table 2: Comparison of computational efficiency between FID, FD-DINOv2 and FWD. FWD exhibits the lowest FLOPs and highest throughput. FD-DINOv has the highest FLOPs and lowest throughput because of its large network, and FID is in between. FLOPs are calculated over individual feature extractors on a single image, and throughput is measured over 50k images.

| Metric | GFLOPs↓ | Throughput (imgs/sec)↑ |
|---|---|---|
| FID | 1.114 | 526 |
| FD-DINOv2 | 15.566 | 53 |
| FWD | **0.006** | **1923** |

Table 3: Evaluation of FID (ImageNet), FID (CelebA) and FWD on the CelebA-HQ and FFHQ datasets. FID (ImageNet) prefers Proj. FastGAN in both datasets, whereas FID retrained on CelebA and FWD both prefer DDGAN in these datasets.

| Dataset | Generator | FID (ImageNet)↓ | FID (CelebA)↓ | FWD↓ |
|---|---|---|---|---|
| CelebA-HQ | Proj. FastGAN | **6.358** | 5.602 | 1.388 |
| | DDGAN | 7.641 | **3.145** | **0.408** |
| FFHQ | Proj. FastGAN | **4.106** | 2.204 | 0.651 |
| | StyleGAN2 | 4.282 | **0.897** | **0.312** |

with the much larger pretrained DINOv2 network. Table 1 lists the resulting distance metrics. For CelebA-HQ and FFHQ, FD-DINOv2 prefers DDGAN and StyleGAN2 images respectively. Here, FD-DINOv2 and FWD agree.

To investigate further, we consider the DNDD-Dataset of agricultural images (Yi et al., 2023) and the Sentinel (Schmitt et al., 2019) dataset. Samples from Proj. FastGAN for DNDD-Dataset and Sentinel are provided in Figures 19 and 21, respectively. Correspondingly, Figures 18 and 20 represent samples from DDGAN for DNDD-Dataset and Sentinel, respectively. In both cases, FID consistently prefers Proj. FastGAN, which was also the case in all prior experiments. Histograms of the InceptionV3 label distribution are depicted in Figure 5b. The histograms indicate domain bias and resemble the observations reported above. On DNDD-Dataset and Sentinel, the verdicts of FD-DINOv2 and FWD are particularly interesting. While both metrics correctly agree on the Sentinel dataset, only FWD correctly prefers DDGAN on the agricultural images.

We carefully chose the DNDD-Dataset, as agriculture images are not commonly used, and the dataset does not resemble ImageNet. We speculate that the LVD-142M dataset may include satellite imagery, contributing to a consistent ranking. Unfortunately, the closed source of the LVD-142M dataset used for training DINOv2 (Oquab et al., 2024) makes it difficult to investigate this domain bias more in detail. In this first set of experiments, we observed that, while FD-DINOv2 provides a partial remedy to the domain bias problems, it still produces an inconsistent ordering for the DNDD-Dataset images. Furthermore, this partial remedy comes at a tremendous computational cost. Table 2 shows that FWD is over 36 times faster to compute than FD-DINOv2.

In a second series of experiments, we investigate the effect of retraining, another expensive solution to the domain bias problem. To this end, we train InceptionV3 on Large-scale Celeb Faces Attributes (CelebA). CelebA comes with 40 facial attributes, which we use to train a classifier. After convergence, we see an exact match ratio of 90% and recalculate FID using this new backbone. The FID (CelebA) column of Table 3 lists the corresponding scores, and FID (CelebA) and FWD provide the same order.

However, in the case of the agricultural dataset, the retrained FID (DNDD) in supplementary Table 10 remains biased, while FWD produces meaningful domain agnostic results. DNDD-Dataset contains 3600 images with seven classes and the task requires detecting nutrient deficiency in winter wheat and winter rye, such as nitrogen, phosphorous, and potassium deficiencies. Once more, we use a retrained InceptionV3 backbone for the FID computation. Compared to CelebA or ImageNet, this is a small dataset and the retrained network does not provide meaningful features. This is an interesting use case since it illustrates that FWD is not just free from data bias. It also provides meaningful feedback for low-resource tasks where retraining InceptionV3 is not feasible.

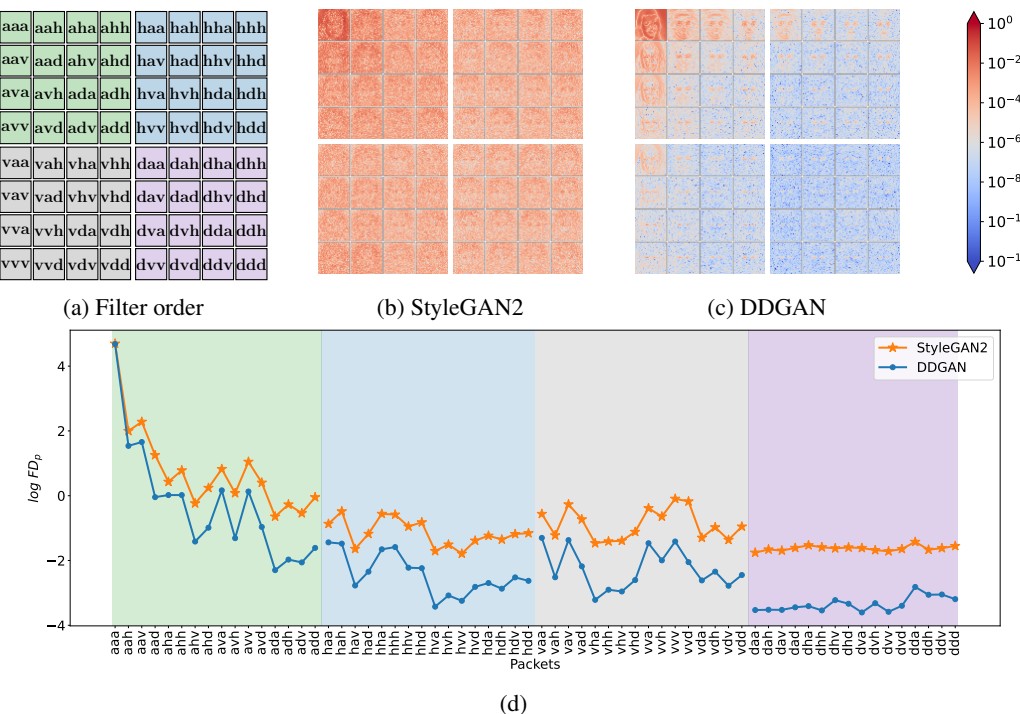

Figure 6: Interpretation of FWD. (a) represents the blueprint for level-3 $\mathcal{W}_p$ transformation. (b) and (c) depict the mean absolute packet difference between CelebA-HQ dataset and generated images by StyleGAN2 and DDGAN, respectively. (d) shows the per-packet Fréchet distances for StyleGAN2 in orange and DDGAN in blue.

In conclusion, experiments in this section indicate that metrics like FID and FD-DINOv2, while helpful, are prone to domain bias when applied to datasets beyond the underlying training datasets. On the contrary, FWD offers a computationally efficient, consistent and domain-agnostic evaluation.

## 4.2 FWD INTERPRETABILITY

A generative metric is interpretable if and only if we can understand the underlying mechanics that produce the ranking. This section explains the decisions made by FWD in one specific case where we focus on samples from DDGAN and StyleGAN2 for CelebA-HQ.

Section 3 formulates FWD as an average of per packet FWD scores. This design choice allows us to understand the overall FWD-score in terms of the individual packet coefficients for each frequency band. Figures 6b and 6c depict the mean absolute difference per packet between the original images of CelebA-HQ and generated samples from StyleGAN2 and DDGAN, respectively. Figure 6d presents both generators' per-packet FWD. Figure 6d shows that DDGAN has a lower Fréchet distance for all packets and averaging the distances over all packages translates into a meaningful metric.

## 4.3 EVALUATION OF ROBUSTNESS

The section follows up on prior work by Kynkäänniemi et al. (2023). The authors generate a large set of samples and find a specific combination of images with an optimal FID. First, the weights of each image are optimized with FID as the objective function. Second, a subset of images is sampled based on the weights. We follow this process and sample 50k images from a large set with optimized weights as probabilities. We employ generated images from StyleGAN2 and real-world images from the FFHQ dataset. Supplementary Table 6 lists the resulting FID, FD-DINOv2 and FWD values. We observe that FWD is robust to FID optimization, whereas FD-DINOv2 showed a little reduction by optimizing FID.

Table 4: Comparing various generative models using Fréchet Wavelet Distance (FWD), Fréchet Inception Distance (FID), DINOv2-Fréchet Distance (FD-DINOv2) and Kernel Inception Distance (KID) on the CelebA-HQ, LSUN-Churches, LSUN-Bedrooms and ImageNet datasets.

| Dataset | Image Size | Method | FID↓ | KID↓ | FD-DINOv2↓ | FWD↓ (ours) |
|---|---|---|---|---|---|---|
| CelebAHQ | 256 | DDIM Song et al. (2021) | 32.333 | 0.0313 | 654.482 | 12.317 |
| | | DDPM (Ho et al., 2020) | 19.101 | 0.0152 | 341.838 | 4.697 |
| | | StyleSwin (Zhang et al., 2022) | 23.257 | 0.0264 | 255.404 | 1.528 |
| | | StyleGAN2 (Karras et al., 2020) | 15.439 | 0.0155 | 593.344 | 0.476 |
| | | DDGAN (Xiao et al., 2022) | **7.203** | **0.0034** | **199.761** | **0.408** |
| Churches | 256 | DDIM (Song et al., 2021) | 11.775 | 0.0043 | 538.400 | 4.919 |
| | | DDPM (Ho et al., 2020) | 9.484 | 0.0036 | 454.402 | 3.546 |
| | | StyleSwin (Zhang et al., 2022) | **3.187** | **0.0005** | **435.967** | 2.835 |
| | | StyleGAN2 (Karras et al., 2020) | 4.309 | 0.0007 | 444.044 | **0.753** |
| Bedrooms | 256 | DDIM (Song et al., 2021) | 25.857 | 0.0094 | 452.419 | 9.521 |
| | | DDPM (Ho et al., 2020) | **16.251** | **0.0058** | **392.481** | **5.187** |
| ImageNet | 64 | Imp. Diff. (VLB) (Nichol & Dhariwal, 2021) | 33.522 | 0.0264 | 670.952 | 2.182 |
| | | EDM (Karras et al., 2024) | 12.295 | 0.0108 | 113.704 | 1.160 |
| | | BigGAN (Brock et al., 2019) | 5.128 | 0.0024 | 170.601 | 0.441 |
| | | Imp. Diff. (Hybrid) (Nichol & Dhariwal, 2021) | **3.091** | **0.0006** | **96.208** | **0.392** |

In addition to FID optimization, we study the impact of image perturbation in supplementary Figure 7. We find that FWD and FD-DINOv2 are closer to a bijective mapping in the presence of perturbation than FID. This behaviour is desirable since we would always expect a larger distance if for example more noise is added. This is not always the case for FID. Consider for example the last quarter of the uniform noise intensity in (b), where FID falls even though more noise is added.

## 4.4 COMPARISON TO STATE OF THE ART

To understand the spectral qualities of existing generative methods for image synthesis, we evaluated various Diffusion and GAN models across a wide range of benchmark datasets.

**Datasets:** We compare common metrics and our FWD on CelebA-HQ (Karras et al., 2018), the Church and Bedroom subsets of the Large-scale Scene UNderstanding (LSUN) dataset (Yu et al., 2015), and finally ImageNet (Russakovsky et al., 2015). In order to retain consistent spatial and frequency characteristics across various image sizes, we use the level 4 packet transform for 256x256 images. For images that are smaller, we use fewer levels, i.e., 3 for 128x128 and 2 for 64x64.

**Generators:** For the evaluation, we use the diffusion approaches Denoising Diffusion Probabilistic Models (DDPM) (Ho et al., 2020), Denoising Diffusion Implicit Models (DDIM) (Song et al., 2021), Improved Diffusion (Nichol & Dhariwal, 2021), DDGAN (Xiao et al., 2022), EDM (Karras et al., 2024), as well as the GAN approaches StyleGAN2 (Karras et al., 2019), StyleSwin (Zhang et al., 2022) and BIGGAN (Brock et al., 2019).

**Hyperparameters:** All generators are evaluated with pretrained weights as provided by the respective paper codebases.

**Metrics:** Table 4 reports the results for FID, KID, FD-DINOv2 and finally our own FWD. FID-scores are obtained by the standard implementation by Seitzer (2020). The ImageNet numbers are computed with 50k images from the validation set. For CelebAHQ and LSUN, we work with 30k images.

Considering CelebA-HQ, FID, KID, FD-DINOv2 and FWD agree most of the time. Considering FID and FWD, only DDPM and StyleSwin are swapped. It is interesting to note that FWD ranks DDIM, DDPM, StyleSwin, and StyleGAN2 on CelebA-HQ and LSUN churches the same way, whereas FID ranks StyleSwin differently on the two datasets. According to FID, StyleSwin performs worse than DDPM and StyleGAN2 on CelebA-HQ but better than these two approaches on LSUN churches. This is counterintuitive, but it can be explained by the domain bias of FID. The supplementary Figure 11a depicts the histograms of top-1 classes classified by InceptionV3 on CelebA-HQ for DDPM and StyleSwin. We observe that DDPM matches the activation histograms of CelebA-HQ more accurately than the histograms of StyleSwin, whereas the histograms of both methods are very similar for LSUN-Church as shown in Figure 11b. As a result, FID ranks StyleSwin worse on CelebA-HQ but better on LSUN-Church. Our metric FWD is not biased by the class distribution and provides a consistent metric for both datasets.

Table 5: Comparison of existing metrics FID, FD-DINOv2 and FWD with Human Error Rate (HER). Higher HER means that participants find the generated images more realistic than the images of the original dataset. HER shows that DDGAN generates perceptually better images.

| Dataset | Generator | FID↓ | FD-DINOv2↓ | FWD↓ | HER(%)↑ |
|---------|-----------|------|------------|------|---------|
| CelebA-HQ | Proj. FastGAN | **6.358** | 685.889 | 1.388 | 20.0 |
| | DDGAN | 7.641 | **199.761** | **0.408** | **32.5** |
| DNDD-Dataset | Proj. FastGAN | **4.675** | **171.625** | 1.442 | 50 |
| | DDGAN | 26.233 | 232.884 | **1.357** | **57** |

We also consider the LSUN-Bedrooms and ImageNet 64 datasets, where FID and FWD agree. We expect pristine performance for FID on ImageNet since this setting is perfectly in its data-domain. Yet, FD-DINOv2 places EDM (Karras et al., 2024) ahead of BigGAN, which is surprising since this does not match with the ranking from FID. FID and FWD agree and arrive at the same ranking.

## 4.5 USER STUDY

To ensure that our metric aligns with human perception, we conduct two types of user studies. The first study demonstrates that FWD does not suffer from domain bias. The second study supports FWD's alignment with human rankings on large-scale diffusion models.

**Datasets and Generators:** In case of the first user study, we use CelebA-HQ and DNDD-Dataset to assess the perceptual quality of images generated by Proj. FastGAN and DDGAN. For the second user study, we use Conceptual Captions (Sharma et al., 2018) as the evaluation dataset and work with pre-trained StableDifusion models, particularly versions 1.5, 2.1, 3.0 (Medium), and 3.5 (Large) from hugging face (https://huggingface.co/stabilityai).

**Results:** Table 5 presents the results of the first user study. A higher Human Error Rate (HER) in the table implies that the participants find the generated images more realistic than the original images. The HER results show that the users identify DDGAN generated images more realistic than Proj. FastGAN generated images. Predominantly, this table highlights FWD's alignment with user preferences across both CelebA-HQ and DNDD-Dataset in comparison to FID and FD-DINOv2. Supplementary Table 7 exhibits the overall alignment of FWD with human perception on large-scale diffusion models. We observe that FWD prefers the latest StableDiffusion-3.5 model over other models, same as the users, whereas FID and FD-DINOv2 rank the StableDiffusion-1.5 model surprisingly better. Moreover, we observe that FWD and other metrics prefer StableDiffusion-1.5 images over 2.1 images. On careful observation of images from these models, we observe that the 2.1 model generates images with artifacts like deformed bodies, extra hands, improper artistic images (like paintings), and some images with white contrast more often than the 1.5 model. We provide the samples from all the StableDiffusion models in Supplementary Section C.1. Overall, the user study demonstrates FWD's alignment with human perception and that it does not suffer from a domain bias.

## 5 CONCLUSION

Modern generative models exhibit frequency biases (Durall et al., 2020), while commonly used metrics such as FID, KID and FD-DINOv2 are affected by domain bias (Kynkäänniemi et al., 2023). To address these limitations, FWD accounts for frequency information without introducing a domain-specific bias. Even though FD-DINOv2 offers a partial solution to this issue, it comes at a very high computational cost and has thus a negative environmental impact. In response, this paper introduced FWD, a novel metric based on the wavelet packet transform. Our metric allows a consistent and domain-agnostic evaluation of generative models, and it is computationally efficient. Our findings show that FWD is robust to input perturbations and interpretable through the analysis of individual frequency bands. FWD in conjunction with traditional metrics ensures a comprehensive and accurate evaluation of generative models while also helping to mitigate domain bias.

ACKNOWLEDGMENTS

This research was supported by the Federal Ministry of Education and Research (BMBF) under grant no. 01IS22094A WEST-AI and 6DHBK1022 BNTrAInee, the Deutsche Forschungsgemeinschaft (DFG, German Research Foundation) GA 1927/9-1 (KI-FOR 5351) and the ERC Consolidator Grant FORHUE (101044724). Prof. Kuehne is supported by BMBF project STCL - 01IS22067. The authors gratefully acknowledge the Gauss Centre for Supercomputing e.V. (www.gauss-centre.eu) for funding this project by providing computing time through the John von Neumann Institute for Computing (NIC) on the GCS Supercomputer JUWELS at Jülich Supercomputing Centre (JSC). The authors heartfully thank all the volunteers who participated in the user study. The sole responsibility for the content of this publication lies with the authors.

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

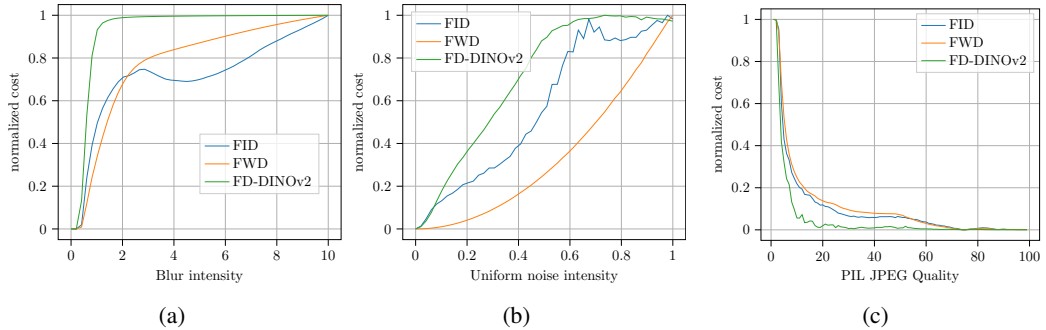

Figure 7: Figures depicting the effect of perturbations such as (a) Gaussian blur, (b) uniform noise corruption and (c) JPEG compression on FID, FWD and FD-DINOv2.

# A    SUPPLEMENTARY

## A.1    ACRONYMS

$\mathcal{W}_p$  Wavelet Packet Transform

**CelebA**  Large-scale Celeb Faces Attributes

**CelebA-HQ**  Large-scale Celeb Faces Attributes High Quality

**CNN**  Convolutional Neural Network

**DDGAN**  Denoising Diffusion GAN

**DDIM**  Denoising Diffusion Implicit Models

**DDPM**  Denoising Diffusion Probabilistic Models

**DNDD-Dataset**  Deep Nutrient Deficiency Dikopshof Dataset

**FD-DINOv2**  DINOv2-Fréchet Distance

**FFHQ**  Flickr Faces High Quality

**FID**  Fréchet Inception Distance

**FWD**  Fréchet Wavelet Distance

**FWT**  Fast Wavelet Transform

**GAN**  Generative Adversarial neural Network

**HER**  Human Error Rate

**IS**  Inception Score

**KID**  Kernel Inception Distance

**LSUN**  Large-scale Scene UNderstanding

**MSE**  Mean Squared Error

**Proj. FastGAN**  Projected Fast GAN

**SWD**  Sliced Wasserstein Distance

**VAE**  Variational AutoEncoder

## A.2    FWD ROBUSTNESS

To supplement Section 4.3, we provide results for FWD's robustness towards various perturbations such as Gaussian blur, uniform noise and JPEG compression in Figure 7. Furthermore, in Table 6, we demonstrate that matching fringe features can be used to optimise FID and FD-DINOv2, whereas FWD does not improve.

Table 6: Matching fringe features for 250k images generated using StyleGAN2 for the FFHQ dataset. By optimizing the sample weights for FID, FD-DINOv2 is also slightly improved. In contrast, FWD penalizes the manipulation of the sample distribution.

| Metric | Random Images | FID-Optimized Images | Change |
|---|---|---|---|
| FID | $4.278 \pm 0.019$ | $\mathbf{2.031 \pm 0.005}$ | -52.53% |
| FD-DINOv2 | $420.223 \pm 0.563$ | $\mathbf{414.048 \pm 0.905}$ | -1.47% |
| FWD | $\mathbf{0.338 \pm 0.017}$ | $0.398 \pm 0.009$ | +17.75% |

### A.3 EXTENDED USER STUDY

For the first user study, following Stein et al. (2023), we presented participants with pairs of real and generated images and asked them to select the realistic image. In this manner, we collected over 1k responses from 50 volunteers. In the second user study, we generated images from the Conceptual Captions (Sharma et al., 2018) validation set and compared our metric with user alignments taken from https://artificialanalysis.ai/text-to-image/arena?tab=Leaderboard.

Table 7: Comparison of metrics FID, FD-DINOv2 and FWD with HER. Higher HER represents a higher prompt alignment percentage according to users. FWD aligns better with HER than FID and FD-DINOv2.

| Generator | FID↓ | FD-DINOv2↓ | FWD↓ | User Rating(%)↑ |
|---|---|---|---|---|
| StableDiffusion-1.5 | $\mathbf{14.904}$ | $\mathbf{124.948}$ | 17.498 | 14 |
| StableDiffusion-2.1 | 15.446 | 132.049 | 21.195 | 22 |
| StableDiffusion-3.0 (Medium) | 18.709 | 158.572 | 6.645 | 45 |
| StableDiffusion-3.5 (Large) | 17.907 | 157.798 | $\mathbf{4.979}$ | $\mathbf{61}$ |

### A.4 THE FAST WAVELET AND WAVELET PACKET TRANSFORMS

This supplementary section summarizes key wavelet facts as a convenience for the reader. See, for example, (Strang & Nguyen, 1996; Mallat, 1999) or (Jensen & la Cour-Harbo, 2001) for excellent detailed introductions to the topic.

The Fast Wavelet Transform (FWT) relies on convolution operations with filter pairs. Figure 8 illustrates the process. The forward or analysis transform works with a low-pass $\mathbf{h}_\mathcal{L}$ and a high-pass filter $\mathbf{h}_\mathcal{H}$. The analysis transform repeatedly convolves with both filters

$$\mathbf{x}_s *_1 \mathbf{h}_k = \mathbf{c}_{k,s+1} \tag{7}$$

with $*_1$ being the 1d-convolution operation, $k \in [\mathcal{L}, \mathcal{H}]$ and $s \in \mathbb{N}_0$, the set of natural numbers. While $\mathbf{x}_0$ is equal to the original input signal $\mathbf{x}$, at higher scales, the FWT uses the low-pass filtered result as input, i.e., $\mathbf{x}_s = \mathbf{c}_{\mathcal{L},s}$ if $s > 0$. The dashed arrow in Figure 8 indicates that we could continue to expand the FWT tree here.

The Wavelet Packet Transform ($\mathcal{W}_p$) additionally expands the high-frequency part of the tree. A comparison of Figures 8 and 9 illustrates this difference. Whole expansion is not the only possible way to construct a wavelet packet tree. See (Jensen & la Cour-Harbo, 2001) for a discussion of other options. In both figures, capital letters denote convolution operators. These may be expressed as Toeplitz matrices (Strang & Nguyen, 1996). The matrix nature of these operators explains the capital boldface notation. Coefficient subscripts record the path that leads to a particular coefficient.

We construct filter quadruples from the original filter pairs to process two-dimensional inputs. The process uses outer products (Vyas et al., 2018):

$$\mathbf{h}_a = \mathbf{h}_\mathcal{L}\mathbf{h}_\mathcal{L}^T, \mathbf{h}_h = \mathbf{h}_\mathcal{L}\mathbf{h}_\mathcal{H}^T, \mathbf{h}_v = \mathbf{h}_\mathcal{H}\mathbf{h}_\mathcal{L}^T, \mathbf{h}_d = \mathbf{h}_\mathcal{H}\mathbf{h}_\mathcal{H}^T \tag{8}$$

With $a$ for approximation, $h$ for horizontal, $v$ for vertical, and $d$ for diagonal (Lee et al., 2019). We can construct a $\mathcal{W}_p$-tree for images with these two-dimensional filters. Figure 10 illustrates the

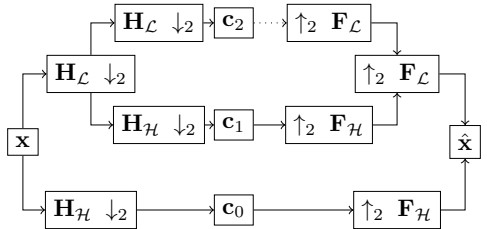

Figure 8: Overview of the Fast Wavelet Transform (FWT) computation. $\mathbf{h}_\mathcal{L}$ denotes the analysis low-pass filter and $\mathbf{h}_\mathcal{H}$ the analysis high pass filter. $\mathbf{f}_\mathcal{L}$ and $\mathbf{f}_\mathcal{H}$ the synthesis filter pair. $\downarrow_2$ denotes downsampling with a factor of two, $\uparrow_2$ means upsampling. The analysis transform relies on stride two convolutions. The synthesis or inverse transform on the right works with stride two transposed convolutions. $\mathbf{H}_k$ and $\mathbf{F}_k$ with $k \in [\mathcal{L}, \mathcal{H}]$ denote the corresponding convolution operators.

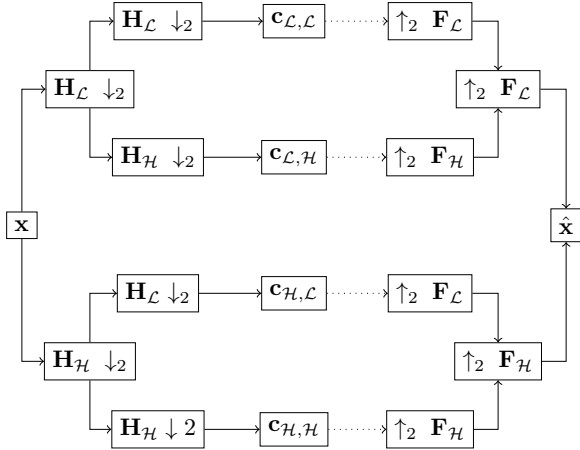

Figure 9: Schematic drawing of the full Wavelet Packet Transform ($\mathcal{W}_p$) in a single dimension. Compared to Figure 8, the high-pass filtered side of the tree is expanded, too.

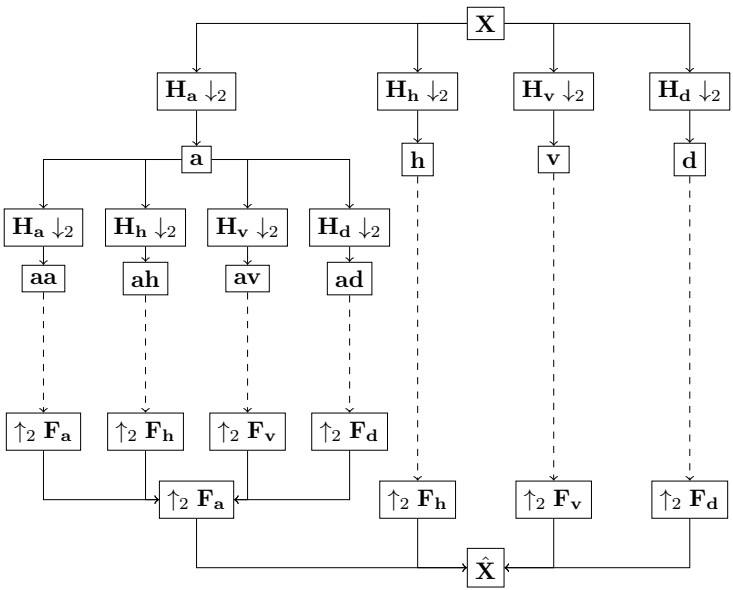

Figure 10: Two dimensional Wavelet Packet Transform ($\mathcal{W}_p$) computation overview. $\mathbf{X}$ and $\hat{\mathbf{X}}$ denote input image and reconstruction respectively. We compute the Fréchet Wavelet Distance (FWD) using the wavelet packet coefficients $\mathbf{p}$. The transform is invertible, the distance computation is therefore based on a lossless representation.

computation of a full two-dimensional wavelet packet tree. More formally, the process initially evaluates

$$\mathbf{x}_0 * \mathbf{h}_j = \mathbf{c}_{j,1} \tag{9}$$

with $\mathbf{x}_0$ equal to an input image $\mathbf{X}$, $j \in [a, h, v, d]$, and $*$ being the two-dimensional convolution. At higher scales, all resulting coefficients from previous scales serve as inputs. The four filters are repeatedly convolved with all outputs to build the full tree. The inverse transforms work analogously. We refer to the standard literature (Jensen & la Cour-Harbo, 2001; Strang & Nguyen, 1996) for an extended discussion.

Compared to the FWT, the high-frequency half of the tree is subdivided into more bins, yielding a fine-grained view of the entire spectrum. We always show analysis and synthesis transforms to stress that all wavelet transforms are lossless. Synthesis transforms reconstruct the original input based on the results from the analysis transform.

## A.5 HISTOGRAM MATCHING - INCEPTIONV3

To understand the results in Table 4 better, we present the histograms of InceptionV3 output labels for images in the datasets CelebA-HQ and LSUN-Church in Figures 11a and 11b, respectively. In both figures, we compare the histograms of the generated images of DDPM and StyleSwin. While StyleSwin generates better images than DDPM, the class distribution of DDPM is closer to the real images compared to StyleSwin on CelebA-HQ. As a result, FID is better for DDPM in Table 4. For LSUN-Church, the distributions are more similar and FID correctly estimates that StyleSwin generates better images than DDPM. In contrast to FID, FWD is not fooled by the class distribution and provides a consistent ranking for DDPM and StyleSwin on both datasets, as reported in Table 4.

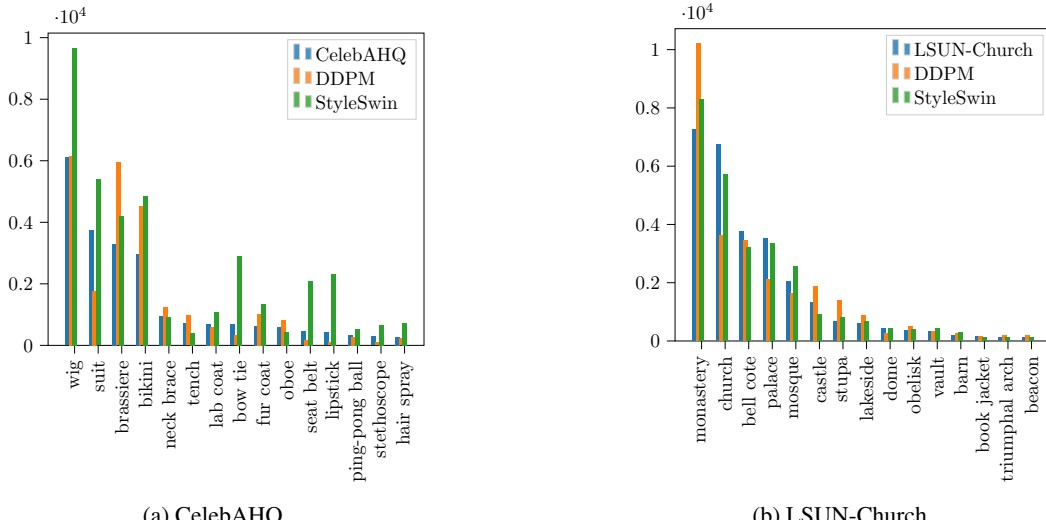

(a) CelebAHQ  (b) LSUN-Church

Figure 11: Histograms of predicted top-1 classes by the InceptionV3 network.

## A.6 COMPUTE DETAILS

While the proposed evaluation metric FWD is very efficient, some of the generative models are expensive. We used 16 nodes with 4 Nvidia A100 GPUs to generate the samples in Table 4.

## A.7 DNDD-DATASET

DNDD-Dataset contains 3600 images with 7 classes and the task requires detecting nutrient deficiency in winter wheat and winter rye, such as nitrogen, phosphorous, and potassium deficiencies. The images were captured over the 2019 growth period at the long-term fertilizer experiment (LTFE) Dikopshof near Bonn and were annotated with seven types of fertilizer treatments. We preprocessed

the dataset by splitting the 1000x1000 resolution image into 256x256 crops. This resulted in 57600 images overall. We trained Proj. FastGAN and DDGAN on this preprocessed dataset.

## A.8 SENTINEL DATASET

The Sentinel dataset consists of 180,662 triplets of Synthetic Aperture Radar (SAR) image patches collected from Sentinel-1 and Sentinel-2 missions. From these, we only use the images from the ROIs_2017_Winter subset, which contain 31,825 images. We train Proj. FastGAN and DDGAN on this subset. The original images are stored in the "tif" format and conversion to "jpg" is made using the official codebase provided by Schmitt et al. (2019).

## A.9 ADDITIONAL METRICS

Here, we present the comparison with additional metrics such as $FID_\infty$ (Chong & Forsyth, 2020), IS (Salimans et al., 2016), $IS_\infty$ (Chong & Forsyth, 2020), Clean-FID (Parmar et al., 2022), and KID (Binkowski et al., 2018). Table 8 extends the results from Table 1. The results show that all the stated metrics suffer from domain bias, as they share the same ImageNet pretrained Inception-V3 backbone.

Table 8: Extended comparison of metrics to detect domain bias. All the metrics which share the pretrained InceptionV3 backbone suffer from domain bias, whereas FWD is domain agnostic.

| | | $FID_\infty\downarrow$ | KID↓ | Clean-FID↓ | IS↑ | $IS_\infty\uparrow$ | FID↓ | FD-DINOv2↓ | FWD↓ |
|---|---|---|---|---|---|---|---|---|---|
| CelebA-HQ | Proj. FastGAN | **6.222** | **0.0020** | **6.729** | **2.925** | **2.545** | **6.358** | 685.889 | 1.388 |
| | DDGAN | 6.961 | 0.0034 | 7.156 | 2.669 | 2.315 | 7.641 | **199.761** | **0.408** |
| FFHQ | Proj. FastGAN | **4.048** | **0.0006** | **4.206** | **5.358** | **3.732** | **4.106** | 593.124 | 0.651 |
| | StyleGAN2 | 4.782 | 0.0011 | 4.597 | 5.307 | 3.714 | 4.282 | **420.273** | **0.312** |
| DNDD-Dataset | Proj. FastGAN | **5.141** | **0.0032** | **5.597** | **2.461** | **2.142** | **4.675** | **171.625** | 1.442 |
| | DDGAN | 25.872 | 0.025 | 26.427 | 2.332 | 2.105 | 26.233 | 232.884 | **1.357** |
| Sentinel | Proj. FastGAN | **5.216** | **0.0030** | **9.087** | **3.846** | 3.257 | **8.96** | 424.898 | 0.755 |
| | DDGAN | 26.154 | 0.0248 | 23.358 | 3.647 | **3.329** | 23.615 | **404.700** | **0.115** |

In addition, Table 9 presents the results of SWD (Karras et al., 2018) and FWD for generated CelebA-HQ images from Proj. FastGAN and DDGAN where we compute each metric five times independently. While SWD is robust to the domain bias, the randomized projections lead to a very high standard deviation, making this metric unreliable in practice. Our proposed metric FWD is deterministic and provides in all runs the same result.

Table 9: Reproducibility of FWD and SWD. We report minimum and mean $\pm$ standard deviation in brackets across 5 independent runs.

| Dataset | Generator | FWD↓ | SWD↓ |
|---|---|---|---|
| CelebA-HQ | Proj. FastGAN | 1.388 (1.388 $\pm$ 0.00) | 169.694 (175.292 $\pm$ 3.54) |
| | DDGAN | **0.408 (0.408 $\pm$ 0.00)** | **99.198 (108.553$\pm$6.81)** |

## A.10 FID PRETRAINED WITH DNDD

As discussed in the results section, fine-tuning the InceptionV3 backbone with DNDD-Dataset does not solve the domain bias problem. Since the dataset consists of only 3600 images, the InceptionV3 network fails to learn representative features to compute FID. Table 10 shows that fine-tuned FID still prefers Proj. FastGAN.

Table 10: Comparison of FID (ImageNet), FID (DNDD) and FWD on DNDD-Dataset. After fine-tuning InceptionV3 on DNDD-Dataset, FID (DNDD) still prefers Proj. FastGAN whereas FWD ranks DDGAN better.

| Dataset | Generator | FID (ImageNet)↓ | FID (DNDD)↓ | FWD↓ |
|---|---|---|---|---|
| DNDD-Dataset | Proj. FastGAN | **4.675** | **20.937** | 1.442 |
| | DDGAN | 26.233 | 52.521 | **1.357** |

## B    EXTENDED DISCUSSION OF RELATED WORK

### B.1    SPECTRAL METHODS

Prior work found neural networks are spectrally biased (Rahaman et al., 2019) and many architectures favor low-frequency content (Durall et al., 2020; Gal et al., 2021; Wolter et al., 2022; Zhang et al., 2022). Related articles rely on the Fourier or Wavelet transform to understand frequency bias. Wavelet transforms, as pioneered by Mallat (1989) and Daubechies (1992), have a solid track record in signal processing. The Fast Wavelet Transform (FWT) and the closely related Wavelet Packet Transform ($\mathcal{W}_p$) are starting to appear more frequently in the deep learning literature. Applications include Convolutional Neural Network (CNN) augmentation (Williams & Li, 2018), style transfer (Yoo et al., 2019), image denoising (Liu et al., 2020; Saragadam et al., 2023), image coloring (Li et al., 2022), face aging (Liu et al., 2019), video enhancement (Wang et al., 2020), face super-resolution (Huang et al., 2017), and generative machine learning (Gal et al., 2021; Guth et al., 2022; Zhang et al., 2022; Phung et al., 2023). Hernandez et al. (2019) use the Fourier transform to measure the quality of human motion forecasting. Zhang et al. (2022) use a FWT to remove artifacts from generated images. Phung et al. (2023) focus on the FWT to increase the inference speed of diffusion models. This work proposes to use the Wavelet Packet Transform ($\mathcal{W}_p$) as an interpretable metric for generators.

### B.2    GENERATIVE ARCHITECTURES

Prior work mainly falls into the three GAN, Diffusion, and Variational AutoEncoder (VAE) architecture groups. The StyleGAN architecture family (Karras et al., 2019; 2020; 2021) is among the pioneering architectures in generative vision. GANs allow rapid generation of high-quality images but suffer from training instability and poor mode coverage (Salimans et al., 2016). Sauer et al. (2021) proposed the Projected Fast GAN (Proj. FastGAN) architecture, which stabilizes and improves training convergence by introducing ImageNet pre-trained weights into the discriminator. The upgraded discriminator pushes the output distribution towards ImageNet. VAE models (Kingma & Welling, 2014; Van Den Oord et al., 2017), on the other hand, enable the generation of diverse image sets, but are unable to produce high-quality images.

Diffusion models (Sohl-Dickstein et al., 2015; Ho et al., 2020; Peebles & Xie, 2023) have emerged as a very promising alternative and produce high-quality images (Ho et al., 2020; Dhariwal & Nichol, 2021) in an autoregressive style. DDPMs, for example, are Markovian processes that learn to gradually separate added noise from data during training. During inference, images are generated from Gaussian noise via a reverse process that requires iterating through all steps to generate an image. Song et al. (2021) reduced the number of sampling steps by introducing DDIM, which relies on a deterministic non-Markovian sampling process. Furthermore, Nichol & Dhariwal (2021) proposed the use of strided sampling, to reduce the sampling timesteps and also provide a performance improvement by using cosine instead of linear sampling. Moreover, Nichol & Dhariwal (2021) adopt a weighted variational lower bound to supplement the Mean Squared Error (MSE) loss. In an attempt to solve the generative learning trilemma (image quality, diversity and fast sampling), Xiao et al. (2022) proposed Denoising Diffusion GAN (DDGAN). The paper parameterizes a conditional GAN for the reverse diffusion process and demonstrates faster generation speed.

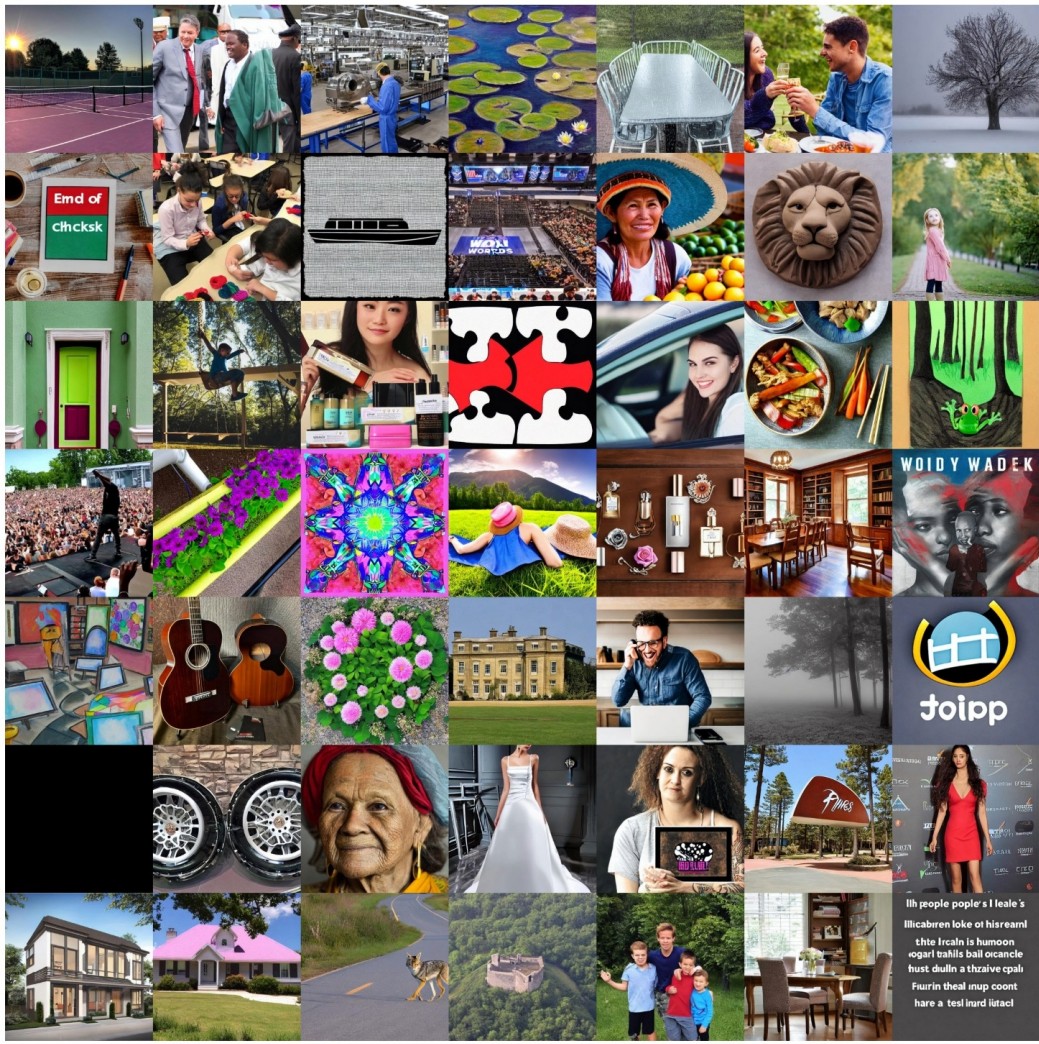

Figure 12: Samples from StableDiffusion v1.5 generated with prompts from the Conceptual Caption validation set.

## C   ADDITIONAL SAMPLES

### C.1   STABLE DIFFUSION

In this section, we provide the generated samples from StableDiffusion models used for user study when evaluated on the Conceptual Captions dataset (Sharma et al., 2018). In particular, we use versions 1.5, 2.1, 3.0 (Medium), and 3.5 (Large), and Figures 12, 13, 14, and 15 represent the samples from these models respectively.

### C.2   DDGAN AND PROJ.FASTGAN

Here we present the additional samples generated from DDGAN and Proj. FastGAN trained on CelebA-HQ, DNDD and Sentinel datasets individually. Figures 16, 17 represent CelebAHQ samples from DDGAN and Proj. FastGAN respectively. Similarly, Figures 18, 19 and Figures 20, 21 depict samples from DNDD and Sentinel datasets, respectively.

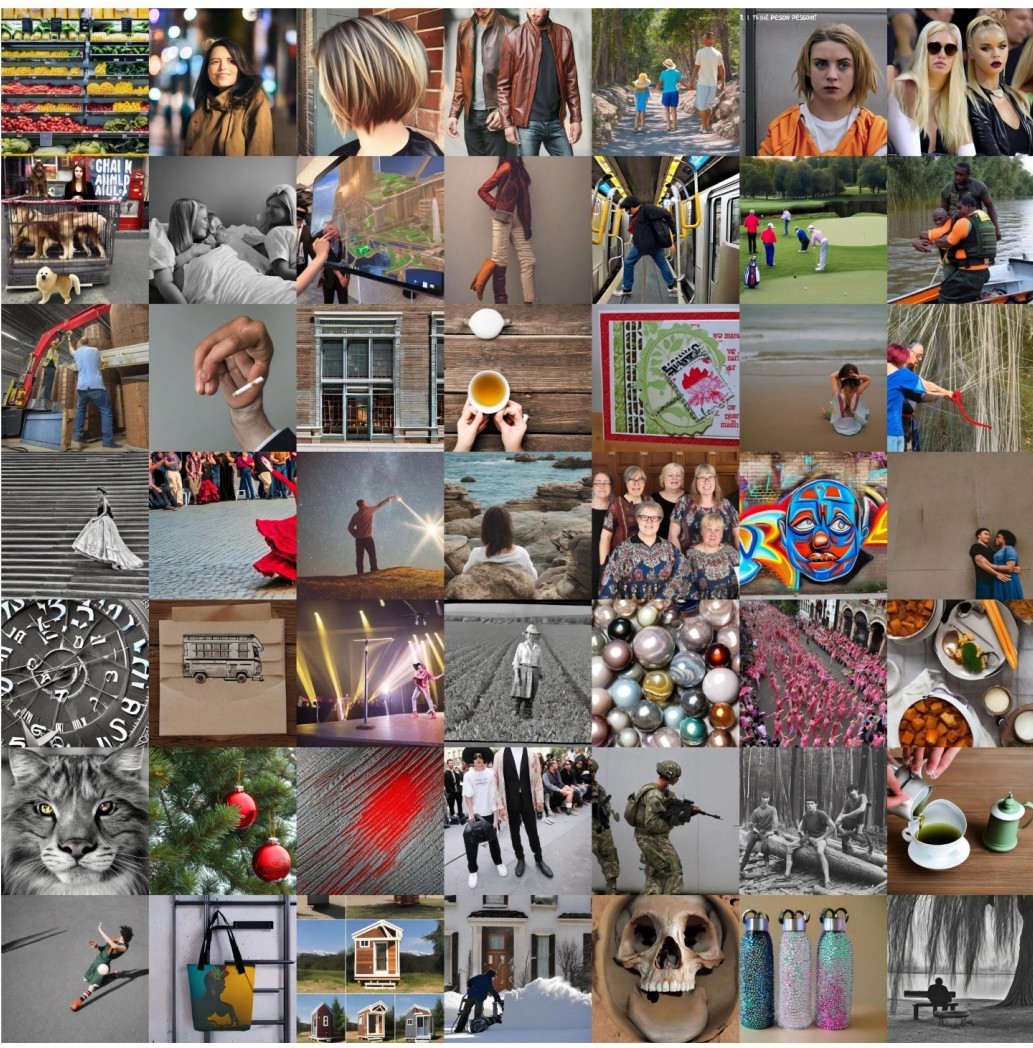

Figure 13: Samples from StableDiffusion v2.1 generated with prompts from the Conceptual Caption validation set.

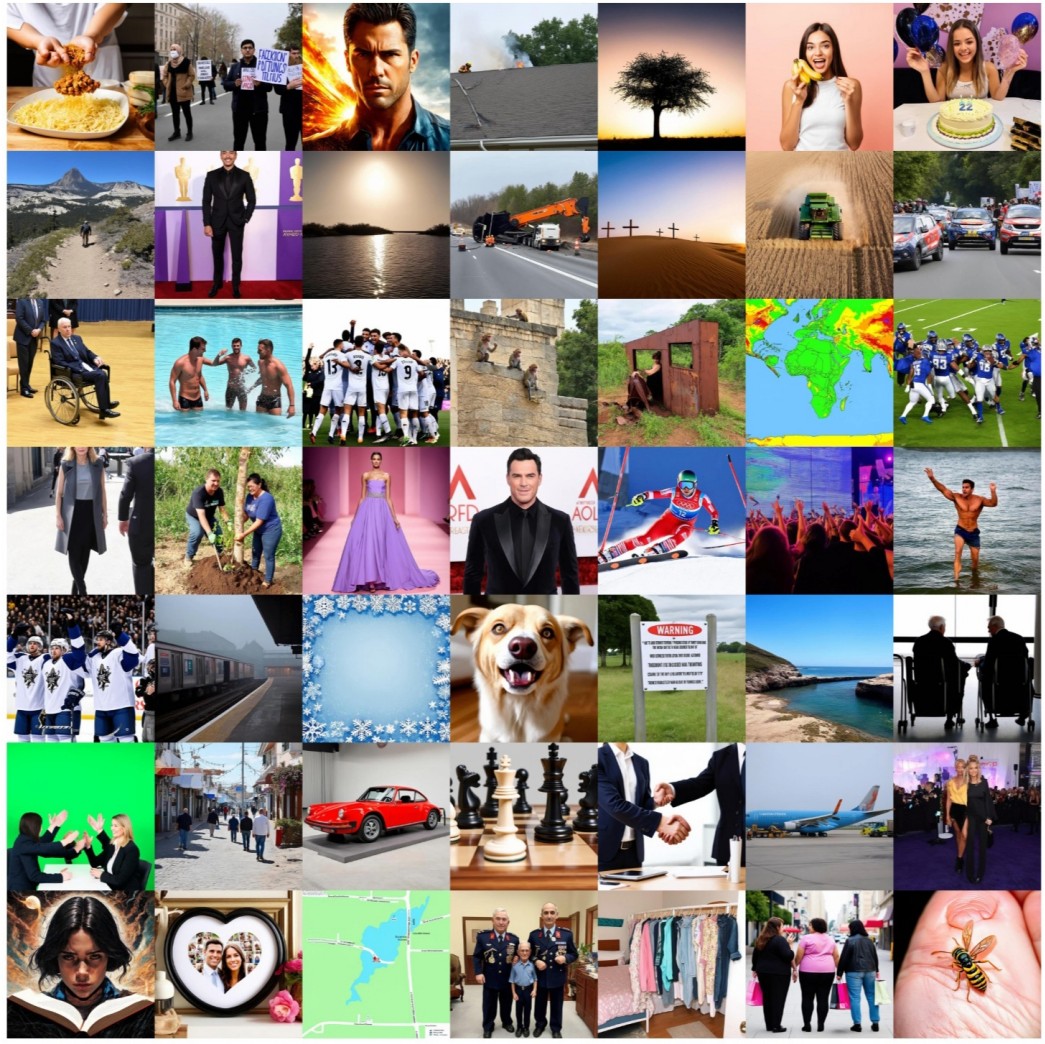

Figure 14: Samples from StableDiffusion v3.0 (Medium) generated with prompts from the Conceptual Caption validation set.

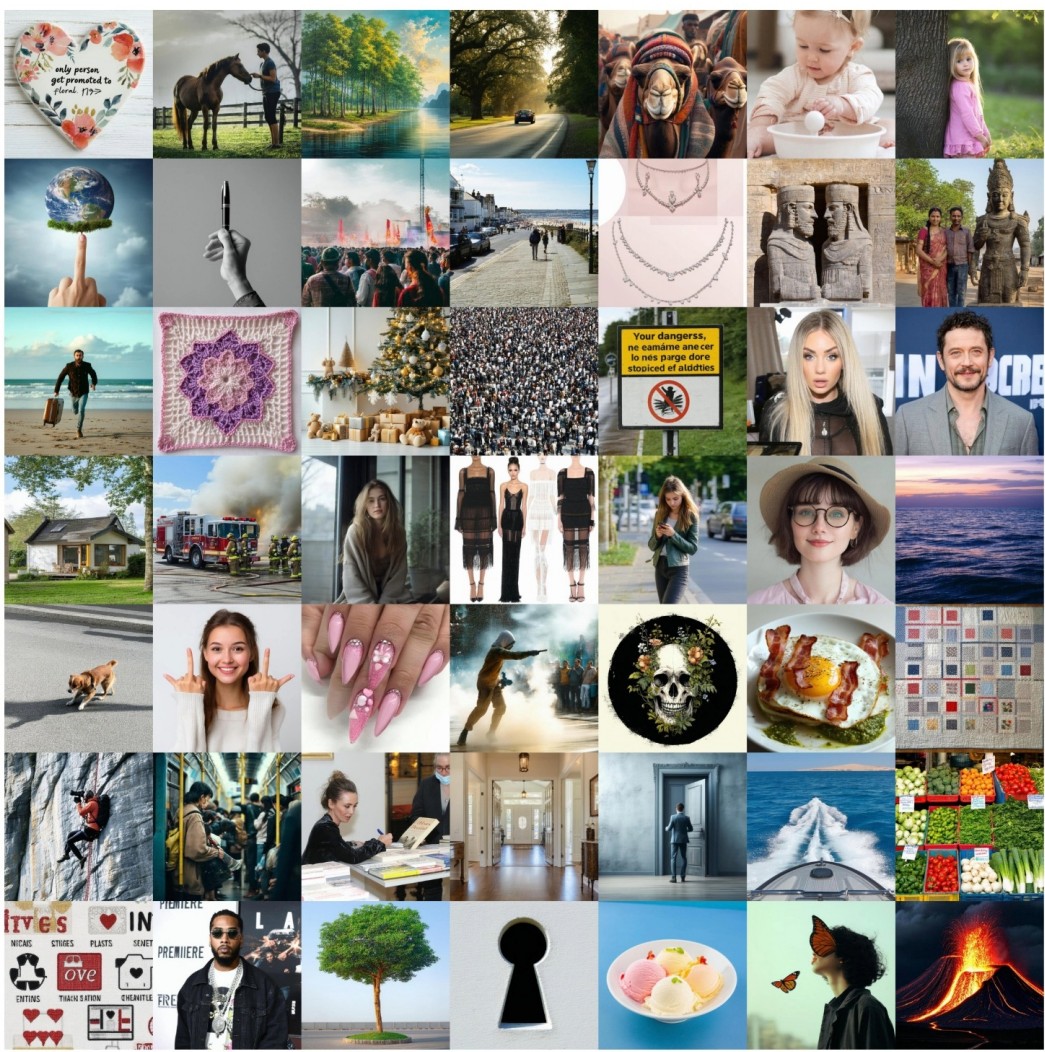

Figure 15: Samples from StableDiffusion v3.5 (Large) generated with prompts from the Conceptual Caption validation set.

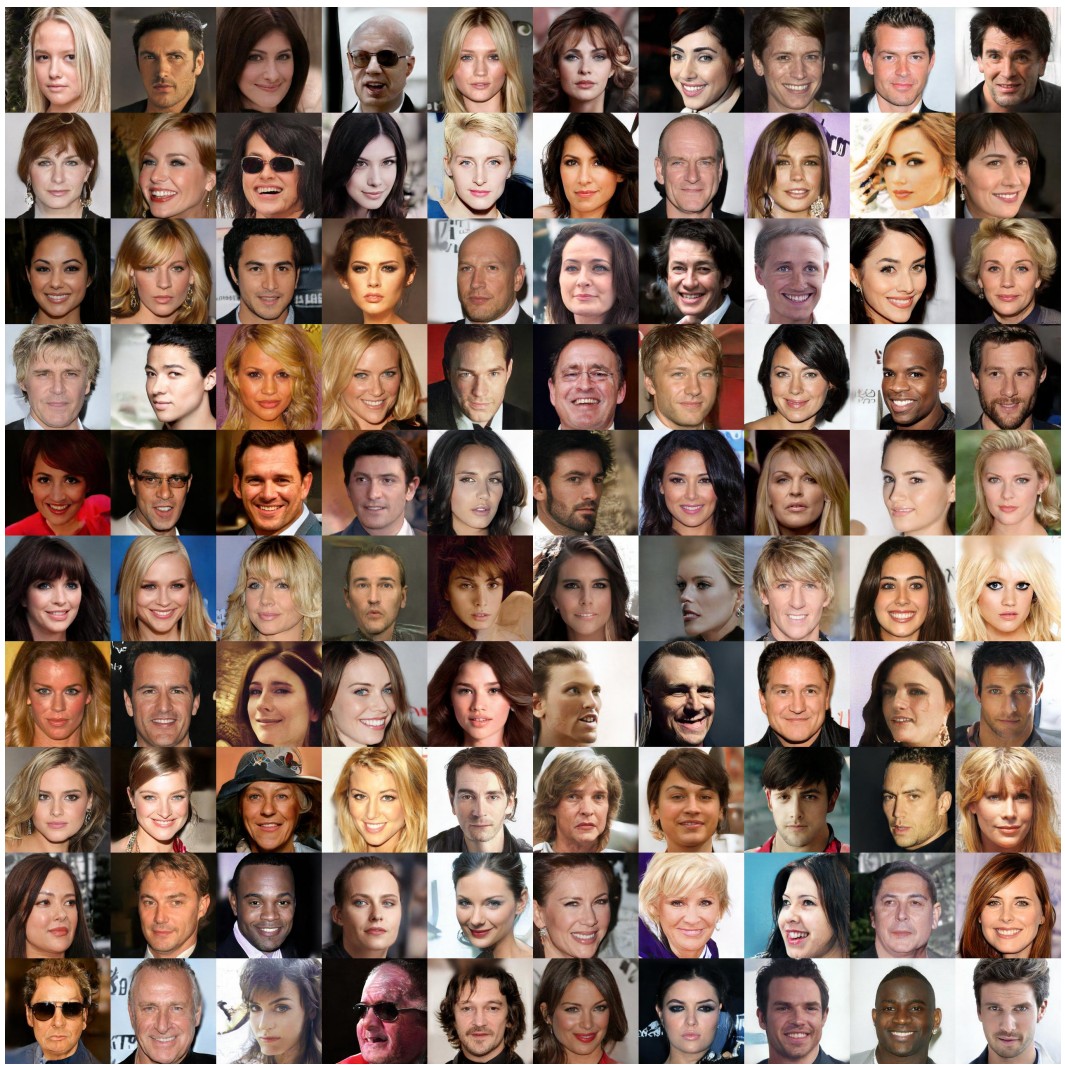

Figure 16: Samples from DDGAN trained on the CelebA-HQ dataset.

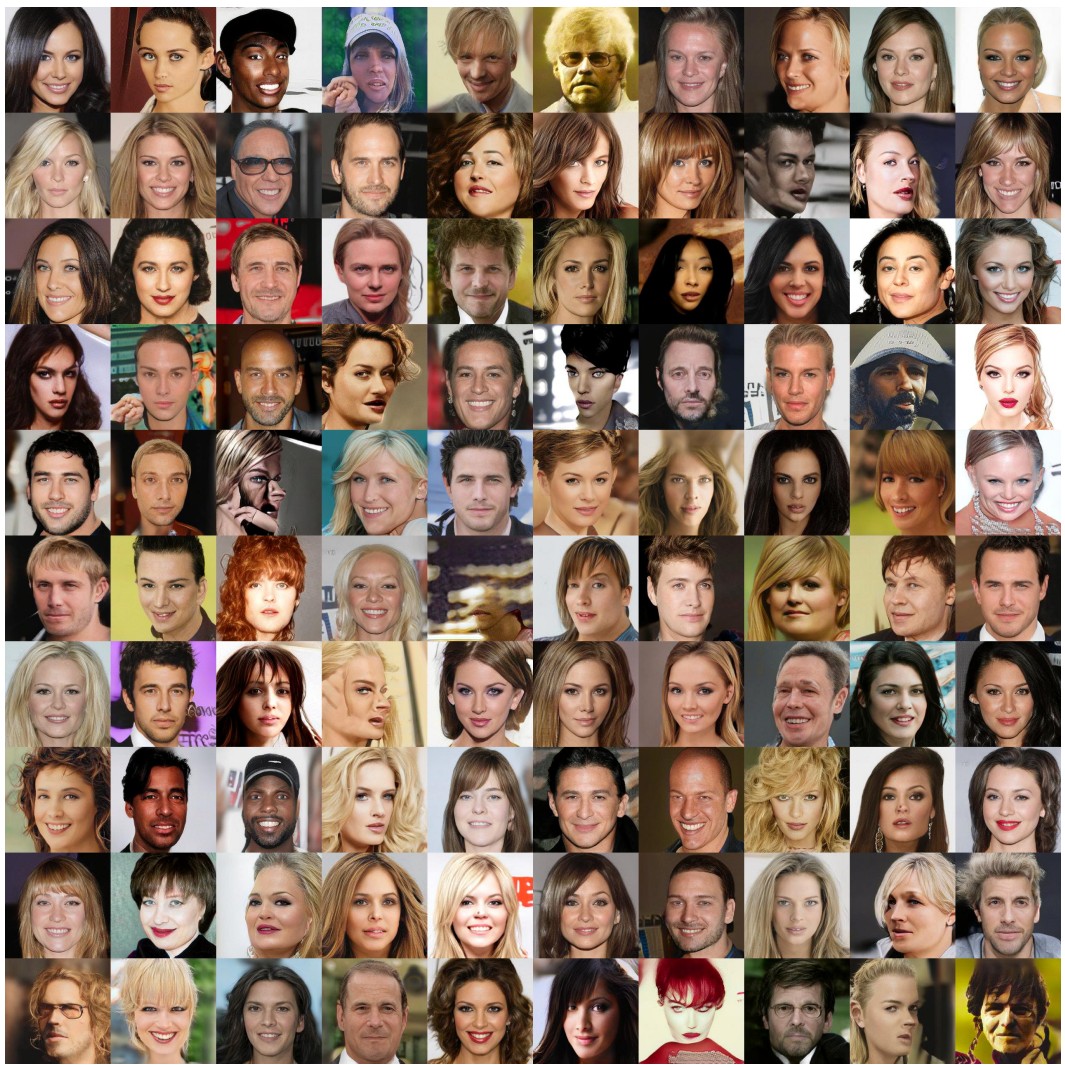

Figure 17: Samples from Proj. FastGAN trained on the CelebA-HQ dataset.

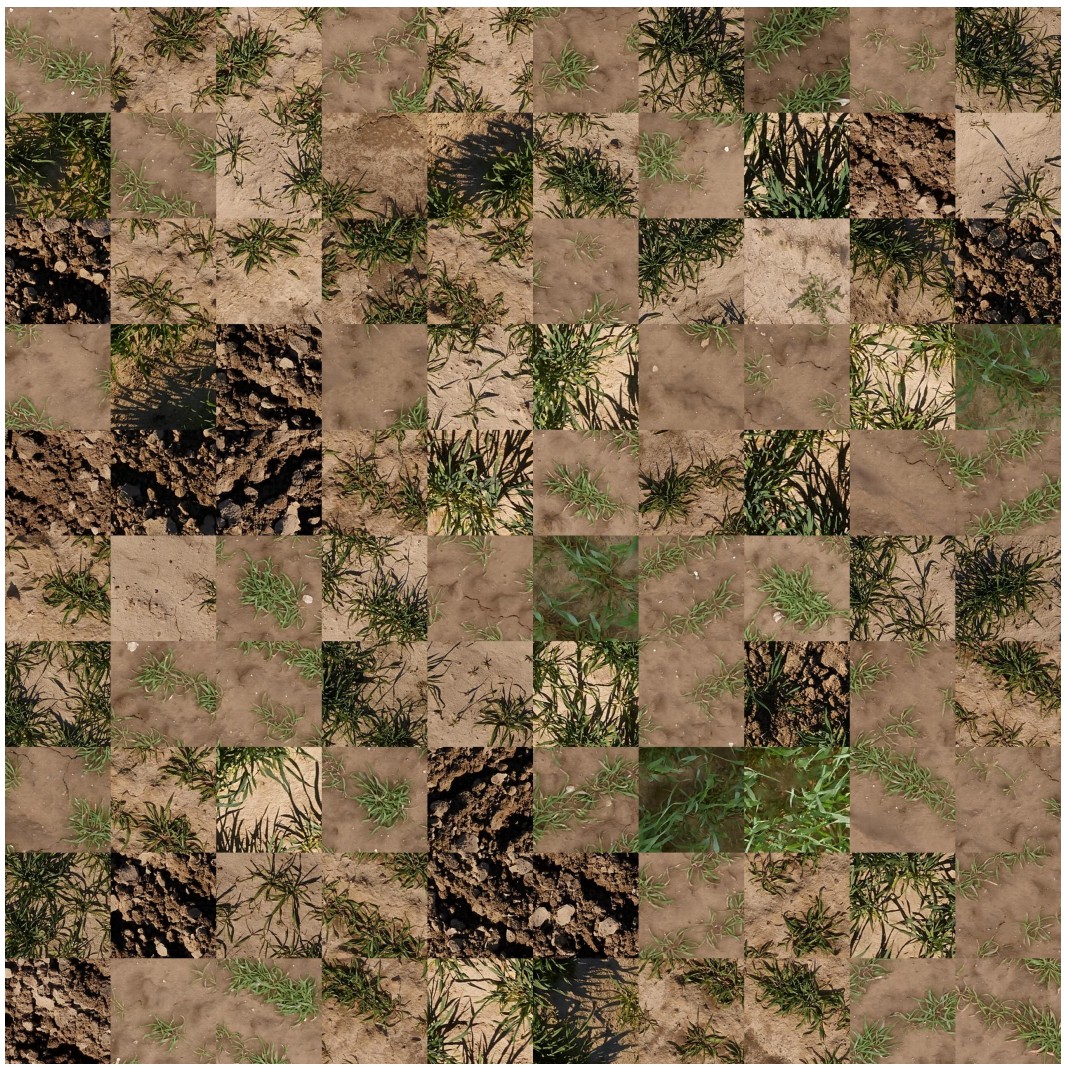

Figure 18: Samples from DDGAN trained on the DNDD-Dataset.

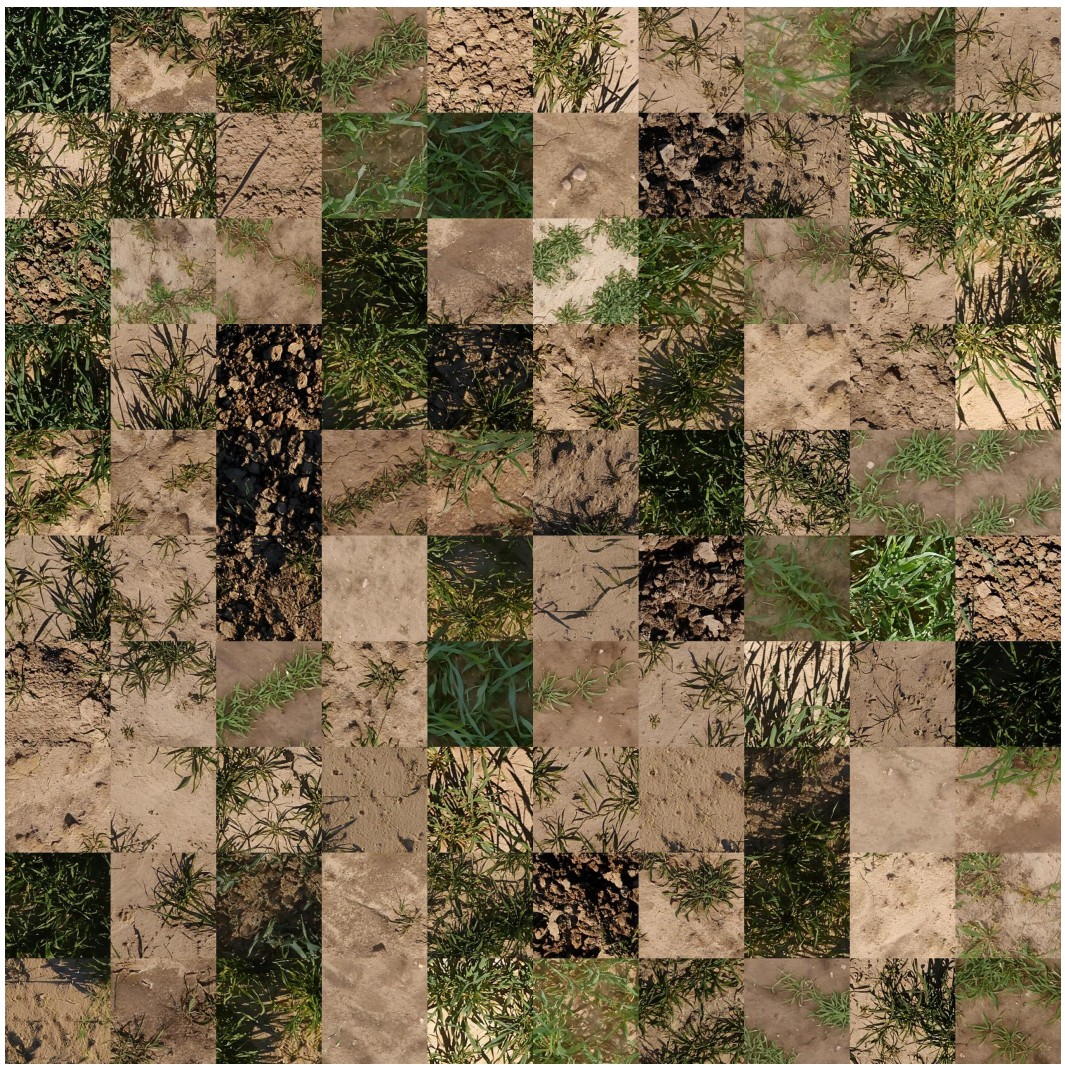

Figure 19: Samples from Proj. FastGAN trained on the DNDD-Dataset.

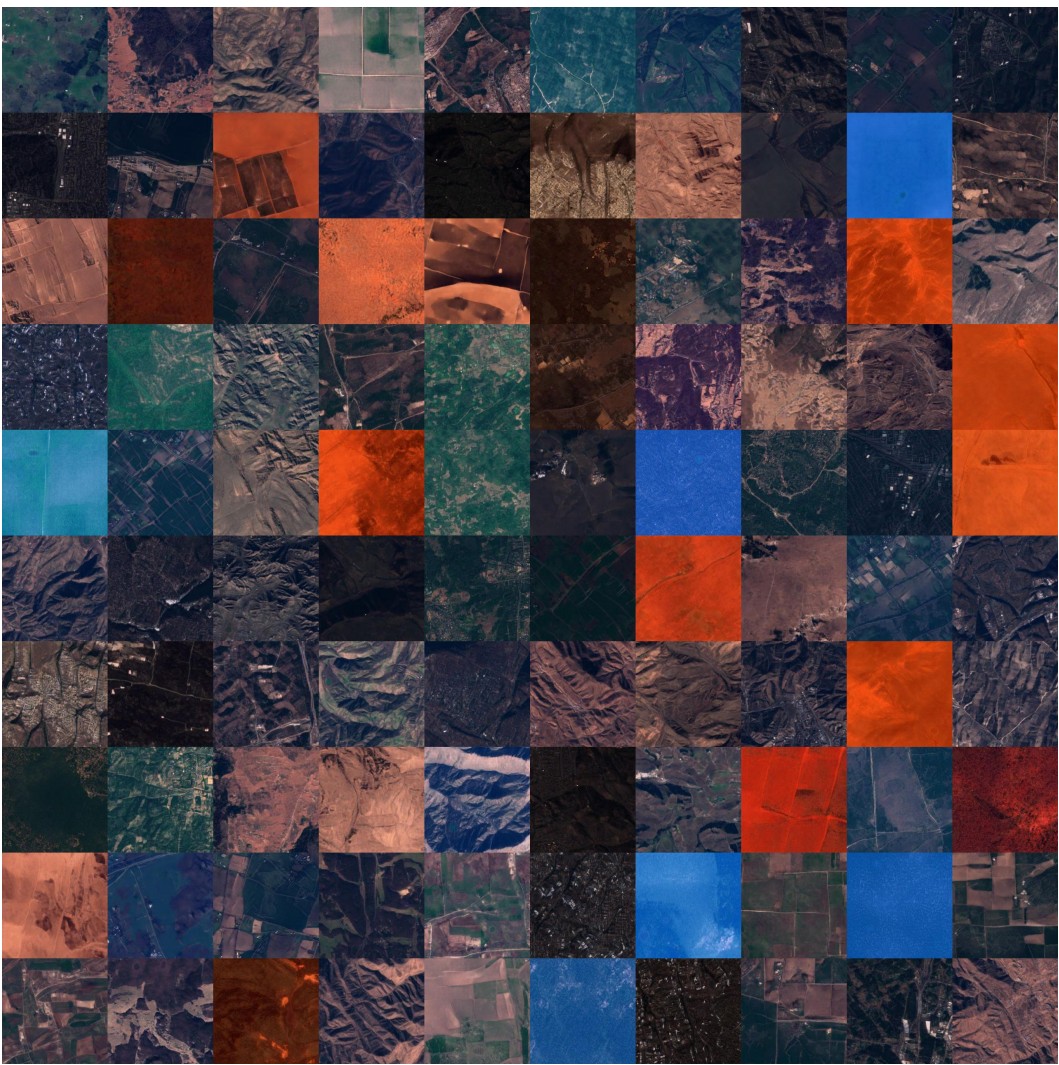

Figure 20: Samples from DDGAN trained on the Sentinel dataset.

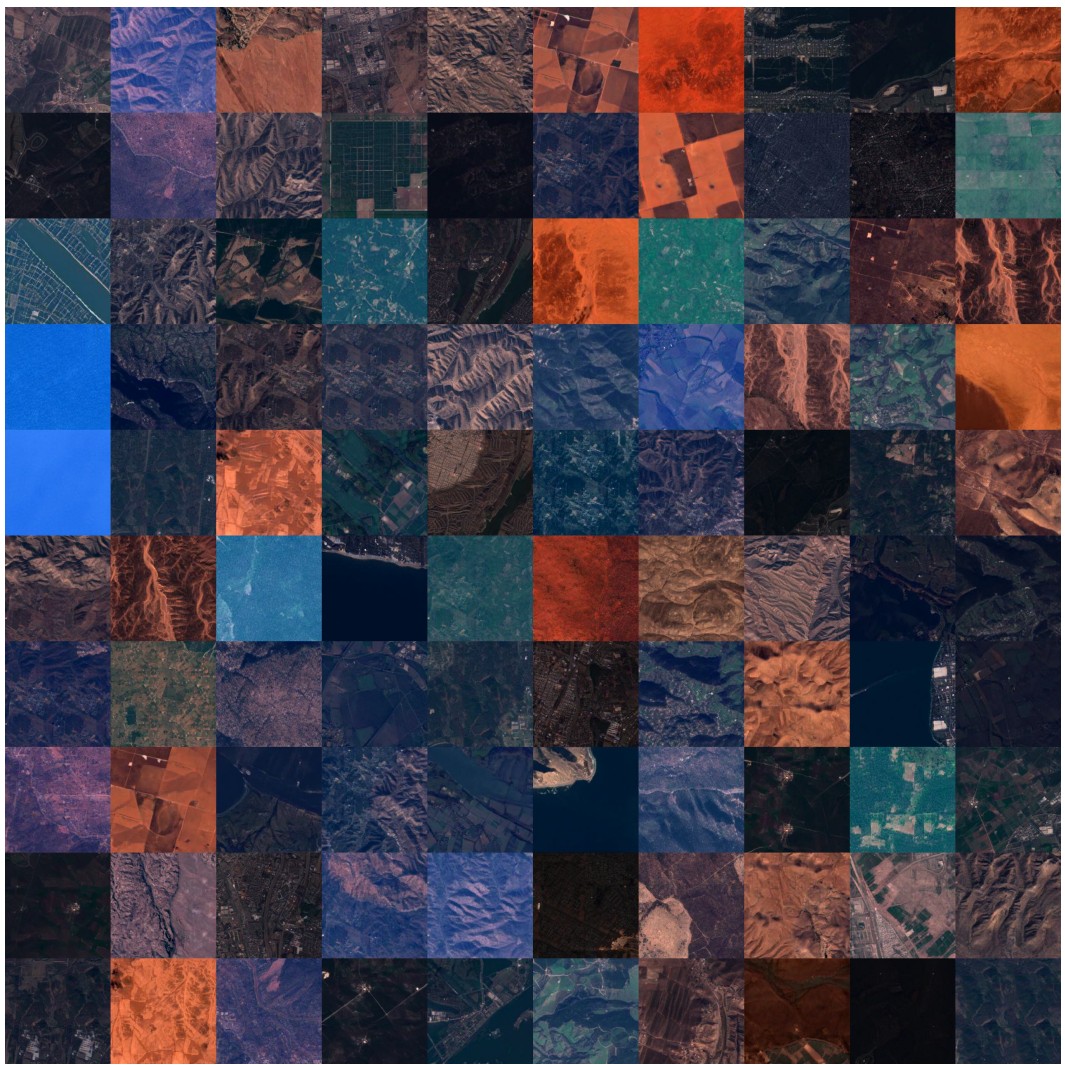

Figure 21: Samples from Proj. FastGAN trained on the Sentinel dataset.

