# OpenReview forum: "Fréchet Wavelet Distance: A Domain-Agnostic Metric for Image Generation"
_ICLR.cc/2025/Conference — ICLR 2025 Poster_

### Official Review · Reviewer_Jm3n · 2024-10-30

**Soundness:** 2
**Presentation:** 2
**Contribution:** 3
**Rating:** 6
**Confidence:** 3

**Summary:**

This paper claims that FID/FD-DINOv2 suffer from the bias of generator and dataset, and proposes the FWD, a domain-agnostic metric based on the Wavelet Packet Transform. The FWD does not rely on any pre-trained network, and thus is general-purpose and dataset-domain agnostic. The author further claims this new metric is more interpretable due to its ability to compute Fréchet distance per packet.

**Strengths:**

1. The proposed solution for enhancing FID is attractive, and the FWD is designed to be general-purpose and dataset-domain agnostic, with a solid foundation.
2. The experiments are well-considered.

**Weaknesses:**

1. The paper appears to be missing comparisons with several competitors. While the FWD is compared with FID/FD-DINOv2, this seems insufficient. For instance, metrics such as IS (Inception Score), KID (Kernel Inception Distance), FID_\infty, IS_\infty, and Clean FID should also be considered. The authors should either include comparisons with these metrics or provide an explanation for their omission. Notably, FID_\infty and IS_\infty are designed to mitigate biases introduced by models, making their inclusion particularly relevant.

2. There is a lack of human evaluation. For a metric evaluating generative models, it would be beneficial to demonstrate alignment with human evaluation results.

3. The underlying mechanism is not entirely clear. Despite the authors’ claim that FWD offers improved transparency, it remains unclear why a hand-designed metric could yield better evaluation results than those based on deep learning techniques. Or you can also explain it.

4. The dataset selection is limited. It appears that all experiments are conducted on just four datasets (face, agricultural, and remote sensing datasets). Including a broader range of datasets, particularly natural images from sources like ImageNet, OpenImage, COCO, or Laion5B, would enhance the robustness of the findings. Or you can also explain it.

[1] Mikołaj Binkowski, Danica J Sutherland, Michael Arbel, and Arthur Gretton. Demystifying mmd gans. arXiv preprint arXiv:1801.01401, 2018.
[2] Min Jin Chong and David Forsyth. Effectively unbiased fid and inception score and where to find them. In Proceedings of the IEEE/CVF conference on computer vision and pattern recognition, pages 6070–6079, 2020.
[3] Gaurav Parmar, Richard Zhang, and Jun-Yan Zhu. On aliased resizing and surprising subtleties in gan evaluation, 2022.

**Questions:**

See weaknesses. As I am not an expert in this field, I would like to adjust my rating if all the reviewers’ concerns are addressed.

---

> ### Author Response · Authors · 2024-11-19
> **We thank Jm3n for taking the time to review our work!**
>
> We address your questions as follows:
>
> _Q: Metrics such as IS (Inception Score), KID (Kernel Inception Distance), $FID_\infty$, $IS_\infty$, and Clean FID should also be considered
>
> We appreciate the reviewer for highlighting additional metrics. Our proposed metric FWD is designed to address domain bias, a critical issue in generative image evaluation as identified by [Kynkäänniemi et al., 2023]. As for the other metrics, we would like to point to the related work section of our paper. In brief, IS was widely used but found to be sensitive to multiple perturbations, does not consider real data distribution [Heusel et al., 2017], and is sensitive to various training runs [Barratt & Sharma, 2018]. Our metric FWD, as well as FID and FD-DINOv2, are robust in these aspects, as illustrated in Figure 7. Secondly, $FID_\infty$ and $IS_\infty$ proposed by [Chong et al., 2019] are built to extrapolate FID and eliminate sample size-related bias. Clean FID [Parmar et al., 2020] studies compression-related artifacts.  [Kynkäänniemi et al., 2023] demonstrated that KID and FID are susceptible to domain bias. Consequently, domain bias remains unaddressed, which is central to our metric.
>
> To provide an analysis, we report our results in the table below. The result demonstrates susceptibility to domain bias. Overall, this highlights the importance of FWD, which is specifically designed to address the domain bias problem in generative models. We believe this extended comparison addresses the reviewer's concern and reinforces the significance of FWD as a robust, fast and computationally efficient metric for tackling domain bias in evaluating generative models.
>
> | | | $FID_\infty$$\downarrow$ | KID$\downarrow$ | Clean-FID$\downarrow$ | IS$\uparrow$ |  $IS_\infty$$\uparrow$ |
> |-|-|-|-|-|-|-|
> | CelebAHQ | Proj. FastGAN |   **6.222**  |**0.0020** | **6.729** | **2.925** |   **2.545**  |
> | | DDGAN | 6.961 | 0.0034 | 7.156 | 2.669 |  2.315 |
> | Sentinel | Proj. FastGAN | **5.216** |**0.0030** | **9.087** | **3.846** | 3.257 |
> | | DDGAN | 26.154 | 0.0248 | 23.358 | 3.647 | **3.329** |
> | DNDD | Proj. FastGAN | **5.141** |**0.0032** | **5.597** | **2.461** | **2.142** |
> | | DDGAN | 25.872 | 0.025 | 26.427 | 2.332 | 2.105 |
> | FFHQ | Proj. FastGAN | **4.048** |**0.0006** | **4.206** | **5.358** | **3.732** |
> | | StyleGAN2 | 4.782 |  0.0011 | 4.597 | 5.307 | 3.714 |
>
> _Q:lack of human evaluation_
>
> We refer to the overall rebuttal response above that addresses this concern.
>
> _Q:The underlying mechanism is not entirely clear_
>
> Interestingly, the Wavelet Packet Transform (WPT) enhances transparency in the following way. The WPT splits the input image into various frequency bands (low -> high ) based on the transformation level, as illustrated in Figure 2. We then compute Frechet Distance (FD) across individual frequency bands, and this individual FD score (Figure 6) provides valuable insights into the model's performance across different frequency bands. For instance, Figure 6 demonstrates that DDGAN achieves a better Frechet Wavelet Distance (FWD) score than StyleGAN2 because DDGAN can generate a better frequency response.
>
> _Q: The dataset selection is limited._
>
> The proposed metric is specifically designed to detect domain biases, and our dataset choices CelebAHQ, FFHQ, DNDD and Sentinel were intentionally selected as they are out-of-domain for ImageNet pretrained metrics like FID. This selection allows us to demonstrate the domain bias in existing metrics effectively. For larger datasets such as LAION and LVD-142M, determining the extent of their in-domain coverage is challenging. For instance, in the case of DINOv2 pre-trained on LVD-142M, we hypothesize that agriculture images are not part of LVD-142M's training set. As a result, FD-DINOv2 produces a biased estimate in this case.
>
> Moreover, no dataset is truly universal, and out-of-domain scenarios will always exist. To identify these, we carefully designed FWD using the WPT, which is independent of the pretraining dataset, delivers domain bias-free estimates, is highly reproducible (refer to sdD9 response), tested (supplementary zip contains code), and is cheap to compute.
>
>
> _References_
>
> [Kynkäänniemi et al, 2023] Tuomas Kynkäänniemi, Tero Karras, Miika Aittala, Timo Aila, and Jaakko Lehtinen. The role of imagenet classes in fréchet inception distance. _ICLR 2023_.
>
> [Barratt & Sharma, 2018] Shane T. Barratt and Rishi Sharma. A note on the inception score. _ICML 2018 workshop.
>
> [Heusel et al, 2017] Martin Heusel, Hubert Ramsauer, Thomas Unterthiner, Bernhard Nessler, and Sepp Hochreiter. Gans trained by a two time-scale update rule converge to a local nash equilibrium. _Neurips 2017_,
>
> [Chong et al, 2020] Chong, Min Jin and Forsyth, David. Effectively Unbiased FID and Inception Score and where to find them. _CVPR_ 2020.
>
> [Parmar et al, 2020] Parmar, Gaurav and Zhang, Richard and Zhu, Jun-Yan. On Aliased Resizing and Surprising Subtleties in GAN Evaluation. _CVPR 2020_.

---

> ### Author Response · Authors · 2024-11-19
> **Extended table with FID, FD-DINOv2, and FWD**
>
> For clarity, we provide the complete table including FID, FD-DINOv2 and FWD here
> | | | $FID_\infty$$\downarrow$ | KID$\downarrow$ | Clean-FID$\downarrow$ | IS$\uparrow$ |  $IS_\infty$$\uparrow$ | FID$\downarrow$ | FD-DINOv2$\downarrow$ | FWD$\downarrow$ |
> |-|-|-|-|-|-|-|-|-|-|
> | CelebAHQ | Proj. FastGAN |   **6.222**  |**0.0020** | **6.729** | **2.925** |   **2.545**  | **6.358** | 685.889 | 1.388 |
> | | DDGAN | 6.961 | 0.0034 | 7.156 | 2.669 |  2.315 | 7.641 | **199.761** | **0.408** |
> | FFHQ | Proj. FastGAN | **4.048** |**0.0006** | **4.206** | **5.358** | **3.732** | **4.106** | 593.124 | 0.651 |
> | | StyleGAN2 | 4.782 |  0.0011 | 4.597 | 5.307 | 3.714 | 4.282 | **420.273** | **0.312** |
> | DNDD | Proj. FastGAN | **5.141** |**0.0032** | **5.597** | **2.461** | **2.142** | **4.675** | **171.625** | 1.442 |
> | | DDGAN | 25.872 | 0.025 | 26.427 | 2.332 | 2.105 | 26.233 | 232.884 | **1.357** |
> | Sentinel | Proj. FastGAN | **5.216** |**0.0030** | **9.087** | **3.846** | 3.257 | **8.96** | 424.898 | 0.755 |
> | | DDGAN | 26.154 | 0.0248 | 23.358 | 3.647 | **3.329** | 23.615 | **404.700**| **0.115** |
>
> Due to space constraints, we cannot include the full table in the rebuttal above.

---

> > ### Comment · Reviewer_Jm3n · 2024-11-24
> >
> > Thanks for the reply which addressed my concerns. I raise my score.

---

### Official Review · Reviewer_whKw · 2024-11-02

**Soundness:** 3
**Presentation:** 3
**Contribution:** 3
**Rating:** 6
**Confidence:** 4

**Summary:**

Modern generative models exhibit frequency biases, while commonly used metrics such as FID , KID and FD-DINOv2 are affected by domain bias.
This paper proposes the Fréchet Wavelet Distance ( FWD ) as a dataset- and domain-agnostic metric for evaluation of generative approaches for image synthesis.
It is  shown that the proposed method is robust to corruption, perturbation, and distractors.
At the same time, its formulation is computationally efficient.

**Strengths:**

The proposed metric Fréchet Wavelet Distance ( FWD ) is simple and has no trained parameters, thus it has high efficiency and is robust to domain, corruption, perturbation, and distractors.

The calculation of FWD is much faster than FID and FD-DINOv2.

On several dataset, FWD produces more reasonable results than FID and FD-DINOv2.

Because of the packet and frequency design, FWD has some interpretability.

**Weaknesses:**

To make the new FWD metric more convincing, a human subject experiment is necessay, as is conducted in the FD-DINOv2 paper. The human evaluation results in FD-DINOv2 paper might be a useful benchmark.

**Questions:**

Many image quality metirc has been proposed for generative model evaluation, but to my knowledge, FID is still the mostly adopted one. I want to hear the authors opinion on this problem. Will evaluating on SOTA generative models like stable diffusion series help the advocation of a new image quality metric?

---

> ### Author Response · Authors · 2024-11-19
> **Thank you for participating in the review process**
>
> Thank you for your questions, we will address these below:
>
> *_make the new FWD metric more convincing, a human subject experiment is necessary_*
>
> We refer to the overall rebuttal response above that addresses this concern.
>
>
> *_Many image quality metrics have been proposed for generative model evaluation, but to my knowledge, FID is still the mostly adopted one. I want to hear the authors opinion on this problem._*
>
> We thank the reviewer for this interesting observation. At the time Frechet Inception Distance (FID) was introduced [Heusel et al, 2017], the prevalent metric for evaluating images was Inception Score (IS). However, FID demonstrated that IS is susceptible to various types of noise and doesn't account for the original data distribution. FID emerged as an alternative to IS during the rise in popularity of GANs, and its utility as a robust metric led to its widespread adoption. While alternatives such as KID, FID_Infinity, and CleanFID have been proposed, FID still remains the metric for benchmarking generative models against prior literature.
>
> This fact forces authors to work with FID for comparisons. Furthermore, IS, KID, FID_Infinity, and CleanFID all share the ImageNet-pre-trained InceptionV3 backbone with FID. With the shared backbone comes a shared domain bias problem. This paper addresses this problem.
>
> This is why our study includes evaluations across well-established GAN models such as StyleGAN2, StyleSwin, BIGGAN, and Proj. FastGAN and recent diffusion architectures like EDM, Improved Diffusion, DDPM and DDIM. Despite its effectiveness, FID has notable limitations, such as susceptibility to domain bias [Kynkäänniemi et al, 2023] and compression algorithms [Parmar et al, 2020]. To address these shortcomings, we propose FWD as an additional metric designed to detect domain bias, which existing metrics fail to capture [Table 1].
>
>
> *_Will evaluating on SOTA generative models like stable diffusion series help the advocation of a new image quality metric?_*
>
> We are working on this point, we will report back once results become available.
>
>
> *_References_*
>
> [Heusel et al, 2017] Martin Heusel, Hubert Ramsauer, Thomas Unterthiner, Bernhard Nessler, and Sepp Hochreiter. Gans trained by a two time-scale update rule converge to a local nash equilibrium. _Advances in neural information processing systems_, 30, 2017a.
>
> [Kynkäänniemi et al, 2023] Tuomas Kynkäänniemi, Tero Karras, Miika Aittala, Timo Aila, and Jaakko Lehtinen. The role of imagenet classes in fréchet inception distance. _ICLR 2023_, Kigali, Rwanda, 2023.
>
> [Parmar et al, 2020] Parmar, Gaurav and Zhang, Richard and Zhu, Jun-Yan. On Aliased Resizing and Surprising Subtleties in GAN Evaluation. _Computer Vision and Pattern Recognition_. 2020.

---

> > ### Comment · Reviewer_whKw · 2024-11-24
> >
> > Thanks for the insightful reply and the human subject experiment, I raise my score. Looking forward to see how those new image quality metric can be applied to SOTA models.

---

> > > ### Author Response · Authors · 2024-11-25
> > > **Preliminary results on Stable Diffusion**
> > >
> > > We thank the reviewer for suggesting this experiment. Using the pretrained HuggingFace models of StableDiffusion2 [Rombach et al, 2022] and StableDiffusionXL [Podell et al, 2024], we evaluated their performance on the MS-COCO validation set. The results, presented in the below table suggest that our metric aligns with FD-DINOv2, both differing from metrics based on InceptionV3 backbone.
> > >
> > > |                   |     FID$\downarrow$    |    KID$\downarrow$    |  FD-DINOv2$\downarrow$  |    FWD$\downarrow$   |
> > > |:-----------------:|:----------:|:---------:|:-----------:|:---------:|
> > > |  StableDiffusion2 |   11.948   |   0.004   | **157.359** | **7.883** |
> > > | StableDiffusionXL | **11.576** | **0.003** |   165.202   |   8.393   |
> > >
> > > This alignment allows us to maintain the argument for the robustness and the relevance of our metric. The observed discrepancy with InceptionV3-based metrics may point to potential ImageNet bias in the SDXL model. Addressing this bias would require a comprehensive analysis of the LAION-5B training set, which is beyond the scope of this work. Overall, the agreement between our metric FWD and FD-DINOv2 emphasizes the reliability of our approach. We believe that these additional experiments support the claims made in the paper and we would greatly appreciate any further consideration of our efforts.
> > >
> > >
> > > *_References_*
> > >
> > > [Rombach et al, 2022] Rombach, Robin, et al. "High-resolution image synthesis with latent diffusion models." Proceedings of the IEEE/CVF conference on computer vision and pattern recognition. 2022.
> > >
> > > [Podell et al, 2024] Podell, Dustin, et al. "Sdxl: Improving latent diffusion models for high-resolution image synthesis." ICLR 2024.

---

### Official Review · Reviewer_sdD9 · 2024-11-03

**Soundness:** 3
**Presentation:** 3
**Contribution:** 3
**Rating:** 6
**Confidence:** 3

**Summary:**

The paper, "Fréchet Wavelet Distance: A Domain-Agnostic Metric for Image Generation", introduces the Fréchet Wavelet Distance (FWD) as a novel metric for evaluating generative models in a domain-agnostic way. FWD leverages the Wavelet Packet Transform (Wp) to project images into a frequency and spatially enriched domain, capturing both the frequency and spatial characteristics of images. This approach enables FWD to evaluate the similarity between generated and real images based on the Fréchet Distance (FD) of their wavelet packet coefficients, providing robustness against domain bias and dataset dependency issues seen in other metrics, such as FID and FD-DINOv2. The authors perform extensive evaluations across various datasets and demonstrate that FWD achieves consistent, domain-agnostic results and outperforms state-of-the-art metrics in terms of robustness and computational efficiency.

**Strengths:**

1. Domain Independence and Robustness: FWD’s reliance on the wavelet transform offers a substantial improvement in domain-agnostic evaluation, avoiding the biases associated with pre-trained models like InceptionV3. This makes it a more universally applicable metric across datasets and model types.
2. Efficient and Scalable: FWD is computationally efficient, with much lower FLOPs compared to FD-DINOv2, making it suitable for large-scale evaluation. The method’s wavelet packet decomposition also enables interpretable results by isolating specific frequency bands for detailed analysis.

**Weaknesses:**

1. Limited Comparison with Alternative Frequency-Based Metrics: Although the paper presents strong results, it lacks comparisons with alternative frequency-based metrics, such as those leveraging Fourier transforms or spectral analysis. This comparison could clarify the specific advantages of wavelet-based decomposition over other spectral methods in generative evaluation.
2. Potential Alignment with Human Evaluation: FID score has been long complaint against that it can not accurately evaluate the quality of generative models, specifically alignment with human preferences. This paper presents some improvement over FID scores on several GAN and Diffusion models, but a wider range of ablations and even human evaluations could better support the validity of this approach.

**Questions:**

Please refer to the weakness part

---

> ### Author Response · Authors · 2024-11-19
> **We thank Reviewer sdD9 for the review.**
>
> We want to address your questions as follows:
>
>
> _Q:Limited Comparison with Alternative Frequency-Based Metrics_
>
> We appreciate the reviewer's interest in alternative frequency-based metrics. Among the current metrics, we are aware of Sliced Wasserstein Distance (SWD) and Multiscale Structural Similarity Index Measure (MS-SSIM) as reported by [Karras et al, 2018]. However, each of these metrics presents unique challenges
> 1. SSIM: SSIM is an effectice pair-wise metric suitable for cases like Super resolution. It isn't relevant when the images are unconditionally generated as the corresponding reference images aren't available.
> 2. SWD: While SWD can detect domain bias, it suffers from significant reproducibiltiy issues, due to the projection onto an independent uniform random basis [Nguyen et al, 2023]. The following table compares the performance of SWD and our proposed metric FWD across five independent runs. SWD demonstrate substantially large standard deviation, whereas FWD remains consistent regardless of number of runs.
>
> |            |                 |        FWD$\downarrow$          |           SWD$\downarrow$           |
> |----------|---------------|-------------------|-----------------------|
> |            |                 |min($\mu \pm \sigma$) |   min($\mu \pm \sigma$)  |
> |  CelebAHQ  |  Proj. FastGAN  |1.388 (1.388$\pm$ 0.00)|169.694 (175.292 $\pm$ 3.54)|
> |            |      DDGAN      |**0.408 (0.408 $\pm$ 0.00)**|**99.198 (108.553 $\pm$ 6.81)**|
>
> Given these limitations, and known bias problem with FID and FD-DINOv2, we believe it is crucial to develop a metric which can solve the domain bias reliably.
> Furthermore, our careful adoption of Wavelet Packet Transform (WPT) not only allows for frequency-band interpretation but also reduces spatial dimensions, simplifying the covariance matrix computation in Frechet Distance.
>
>
> _Q: Potential Alignment with Human Evaluation_
>
> We refer to the overall rebuttal response above that addresses this concern.
>
> _References_
>
> [Nguyen et al, 2023] Nguyen, Khai, Tongzheng Ren, and Nhat Ho. "Markovian sliced Wasserstein distances: Beyond independent projections." _Advances in Neural Information Processing Systems 36 (2023)_.
>
> [Karras et al, 2018] Progressive Growing of GANs for Improved Quality, Stability, and Variation, _ICLR 2018_.

---

> > ### Author Response · Authors · 2024-11-25
> > **Follow up on rebuttal feedback**
> >
> > As we approach the end of the discussion phase, we wanted to follow up regarding our responses to your valuable comments. We have carefully reviewed and addressed each of the points raised and hope that our responses effectively demonstrate the merits of our paper. We truly appreciate your dedicated time and effort in reviewing our work. Understanding that your schedule may be busy, we greatly value any additional feedback on our rebuttal or confirmation that our responses have resolved your concerns. If there are any remaining questions or comments, we would be happy to address them in the mean time. Otherwise, we kindly request that you consider adjusting your rating.
> >
> > Thank you again for your thoughtful and constructive review.

---

> > ### Comment · Reviewer_sdD9 · 2024-11-26
> >
> > Dear authors,
> >
> > >Re: Limited Comparison with Alternative Frequency-Based Metrics
> >
> > Thanks for providing your reasoning, I have no further questions regarding this part
> >
> > > Re: Human preference
> >
> > Thanks for providing additional metrics regarding human preference score on GAN models in the rebuttal responses. I am aligned with reviewer whKw and interested in how proposed score matches human preferences on SoTA diffusion models. Specifically, does the score matches typical human perceived rankings. Can we perform ablations on SD1.5, SD2.1, SD3 and SD3.5 (i.e. to validate if it will align with https://artificialanalysis.ai/text-to-image/arena?tab=Leaderboard)
> >
> > Best Regards,

---

> > > ### Author Response · Authors · 2024-11-29
> > > **Stable Diffusion results**
> > >
> > > Thank you for suggesting the additional ablation. We used the default implementation setting from HuggingFace and generated images with 30 inference steps. Additionally, for SD1.5 around 200 images were flagged as NSFW by the model, and they are removed. For FID and FD-DINOv2, images were resized to the resolution required by these metrics. FWD was evaluated with the original resolution of 512 by 512 pixels using a level five wavelet packet transformation. This follows the guideline in our paper, which specifies that "we reduce the transformation level by a factor of 1 for a reduction in image size by half and vice-versa". All models were evaluated on the Conceptual Captions (CC) validation set because of its manageable size, which fits our hardware given large-scale models.
> > >
> > > Below, we present the evaluation results for the specified StableDiffusion (SD) models:
> > >
> > >
> > > |   Generator     |   FID $\downarrow$  | FD-DINOv2 $\downarrow$ |   FWD  $\downarrow$  | Users (prompt alignment) $\uparrow$ |
> > > |-----------------|---------------------|------------------------|----------------------|-------------------------------------|
> > > |      SD 1.5     | **14.904**          |  **124.948**          | 17.498               |            14%                      |
> > > |      SD 2.1     | 15.446              |  132.049               | 21.195               |            22%                      |
> > > | SD 3.0 (Medium) | 18.709              |  158.572               |  6.645               |            45%                      |
> > > |  SD 3.5 (Large) | 17.907              |  157.798               |  **4.979**          |            **61%**                 |
> > >
> > > We observe that FWD ranks SD-3.5 highest among the four models, followed by SD-3, SD-1.5 and SD-2.1 in terms of image quality. Conversely, FID and FD-DINOv2 metrics prefer SD-1.5. Please note that the leaderboard at https://artificialanalysis.ai/text-to-image/arena?tab=Leaderboard focuses on prompt-to-image alignment, where users are given a prompt and asked to rate text-to-image agreement. By design, FWD doesn't take text into account. The CC dataset contains image and text pairs. To answer your question we compute the distance between the CC images and images generated using the corresponding text as input-prompt to the SD-Models [Esser, Patrick, et al].
> > >
> > > Given a prompt, users prefer SD 3.5 images over other SD models. According to FWD, SD 3.5 produces the lowest distance between generated images and the original images corresponding to the prompt. We observe that SD1.5 is ranked better than SD2.0 by all three metrics. We observed SD2.1 generates images with artifacts like deformed bodies, extra hands, improper artistic images (like paintings), and some images with white contrast more often than SD1.5. We provide samples for SD1.5 and SD2.1 models as a context in these anonymized links (https://i.ibb.co/c1YjNKv/A1.png, https://i.ibb.co/ZcjFkf6/A2.png).
> > >
> > > Finally, our paper presents evidence supporting FWD's ability to detect domain basis. The reviewer's question moves in a new direction. To answer this, we provide this additional experiment, which demonstrates that FWD agrees with the provided user score. We believe this experiment further strengthens the robustness of our metric, and we will include it in the camera-ready version.
> > >
> > > **References**
> > > Esser, Patrick, et al. "Scaling rectified flow transformers for high-resolution image synthesis." Forty-first International Conference on Machine Learning. 2024.

---

> > > > ### Comment · Reviewer_sdD9 · 2024-12-02
> > > >
> > > > Dear authors,
> > > >
> > > > Thanks for your detailed ablation study, this answers all my concerns and I am convinced that FWD indeed can be used as a better metric than FID for image generation tasks. With this being said, I've raised the presentation score but will maintain the overall score as 6.
> > > >
> > > > Best regards

---

### Author Response · Authors · 2024-11-19
**Overall rebuttal response**

We appreciate the reviewers' constructive comments and valuable feedback. We are delighted to read that reviewers found our work novel (sdD9), efficient (sdD9, whKW), domain-agnostic (sdD9, Jm3n, whKw), robust (sdD9, whKW), interpretable (whKw, sdD9), with a solid foundation (Jm3n) and sound experimentation (Jm3n). Naturally, our reviewers also voice some concerns. We address the key points here.

First, we clarify the effect of pre-trained datasets (ImageNet) in the evaluation metrics of generative models. Generators utilizing ImageNet backbones often introduce unintended ImageNet-specific bias into the evaluation. Consistent with prior work [Kynkäänniemi et al., 2023], we confirm the domain bias in evaluating with FID and further show it also affects FD-DINOv2 (Table-1). To address this issue, we propose Frechet Wavelet Distance (FWD), a novel metric grounded by mathematical foundations of Wavelet Packet Transform (WPT). We demonstrate that FWD is free from domain bias problems.

To further substantiate our claims and also address the key concerns of reviewers, we conducted a user study involving around 50 participants and collected approximately 1000 responses. Following the user study methodology [George et al, 2024], participants were asked to select the more realistic image from a pair consisting of the generated image and the original dataset image. This study was conducted and deployed using Streamlit and Google Cloud Platform. We included the user responses and images in zip format in the paper's supplementary material as evidence.
The table below lists the results along with the Human Error Rates (HER) (higher is better) for each model and corresponding dataset.
||Generator|FID$\downarrow$|FD-DINOv2$\downarrow$|FWD(ours)$\downarrow$|HER$\uparrow$|
|-|-|-|-|-|-|
|CelebAHQ|Proj. FastGAN|**6.358**|685.889|1.388|20.0|
||DDGAN|7.641|**199.761**|**0.408**|**32.5**|
|DNDD|Proj. FastGAN |**4.675**|**171.625**|1.442|50.0|
||DDGAN|26.233|232.884|**1.357**|**57.0**|

The human evaluation asserts the effectiveness of FWD in detecting domain bias. Participants were able to identify unrealistic images generated by Proj. FastGAN at a higher rate than those from DDGAN, this is consistent with FWD's quantitative results across both CelebAHQ and DNDD datasets. In conclusion, our experiments suggest that FWD offers a computationally efficient, fast and reproducible metric and effectively detects domain bias, a prevalent issue in existing metrics. The combination of empirical validation and human evaluation results ensures that FWD addresses a critical gap in generative evaluation metrics.  We believe this work will serve a greater interest in the community, and we will incorporate this user study discussion into the paper at the end of the discussion period.

We hope the rebuttal has sufficiently addressed everyone's concerns. If that's indeed the case at the end of the discussion period, please consider raising your score.


*_References_*

[Kynkäänniemi et al, 2023] Tuomas Kynkäänniemi, Tero Karras, Miika Aittala, Timo Aila, and Jaakko Lehtinen. The role of imagenet classes in fréchet inception distance. _ICLR 2023_.

[George et al., 2024] Stein, George, et al. "Exposing flaws of generative model evaluation metrics and their unfair treatment of diffusion models." _NeurIPS 2024_.

---

> ### Author Response · Authors · 2024-11-27
>
> We sincerely thank the reviewers for their feedback and active participation in the discussion. We have updated the paper to include results from the user study and additional metrics in the supplementary material, adhering to the page limit regulations set by ICLR. Regarding the stable diffusion experiments, for reviewer whKw, we so far provide results for Stable Diffusion 2 and Stable Diffusion XL. As the additional experiments requested by sdD9 involve generating images using other stable diffusion models, which require more time to complete, we plan to include these results in the camera-ready version and will notify you once the results become are available. We hope that, based on the discussion so far, we could convince you that our paper will effectively address those open points. Thank you for your time and consideration.

---

### Meta-Review · Area_Chair_PJLE · 2024-12-19

**Metareview:**

Summary: The paper presents FWD as a novel, domain-agnostic metric for evaluating images in generative models. It uses the Wavelet Packet Transform and measures similarity between generated and real images based on the Frechet Distance of the wavelet coefficients, effectively addressing the domain bias problem. The proposed FWD metric is computationally efficient as shown by evaluations on various datasets.

Strength: FWD is a simple, parameter-free metric, efficient to compute metric. It’s also shown to be robust to domain variations. The experimental results support the paper claims

Weakness: The scale of the experiments is rather smaller. Can this be directly used to re-evaluate text-to-image generation leaderboards? Human evaluation is provided only after requested by the reviewers. I encourage authors to include the human evaluation in the main body of the paper, rather than the supplementary.

Acceptance Reason: It addresses an important problem in image quality evaluation in generative models. FID is studied to be susceptible to biases in the generator or the dataset. This work explores a domain-agnostic and efficient metric, which will benefit the community as an alternative metric, and inspire future research in studying related-metrics at a larger scale.

**Additional Comments On Reviewer Discussion:**

The paper received 3x marginally above acceptance threshold. The work is solid, well motivated, nicely written and supported by sound experiments. Reviewers raised a number of points. Almost all concerns of reviewers are addressed by the authors as acknowledged by the reviewers. There is a common question on the alignment of FWD with human evaluation. The rebuttal provided convincing results showing FWD in agreement with user scores on the Stable Diffusion family of models. I encourage authors to add these results into the main paper.

---

### Decision · Program_Chairs · 2025-01-22

Accept (Poster)